



# Imprint of chaotic ocean variability on transports in the Southwest Pacific at interannual timescales

Sophie Cravatte[1], Guillaume Serazin[2], Thierry Penduff[3], Christophe Menkes[4]

1- LEGOS, Université de Toulouse, IRD, CNES, CNRS, UPS, Toulouse, France
2- Climate Change Research Center, University of New South Wales, Sydney, Australia
3- Université Grenoble Alpes, CNRS, IRD, Grenoble-INP, Institut des Géosciences de l'Environnement (IGE), Grenoble, France
4- ENTROPIE, IRD, CNRS, UR, UNC, Ifremer, Nouméa, New Caledonia

*Correspondence to*: Sophie Cravatte (sophie.cravatte@ird.fr)


**Abstract**

The Southwest Pacific Ocean sits at a bifurcation where southern subtropical waters are redistributed equatorward
and poleward by different ocean currents. The processes governing the interannual variability of these currents are
not completely understood. This issue is investigated using a probabilistic modeling strategy that allows
disentangling the atmospherically-forced deterministic ocean variability and the chaotic  intrinsic ocean variability.
A large ensemble of 50 simulations performed with the same ocean general circulation model (OGCM) driven by
the same realistic atmospheric forcing that only differ by a small initial perturbation is analyzed over 1980-2015.
Our results show that, in the Southwest Pacific, the interannual variability of the transports is strongly dominated
by chaotic ocean variability south of 20°S. In the tropics, while the interannual variability of transports and eddy
kinetic energy modulation is largely deterministic and explained by El Nino Southern Oscillation (ENSO), ocean
nonlinear processes still explain 10 to 20% of their interannual variance at large-scale. Regions of strong chaotic
variance generally coincide with regions of high mesoscale activity, suggesting that a spontaneous inverse cascade
is at work from mesoscale toward lower frequencies and larger scales. The spatiotemporal features of the low-
frequency oceanic chaotic variability are complex but spatially coherent within certain regions. In the Subtropical
Countercurrent area, they appear as interannually-varying, zonally elongated alternating current structures, while
in the EAC region, they are eddy-shaped. Given this strong imprint of large-scale chaotic oceanic fluctuations, our
results question the attribution of interannual variability to the atmospheric forcing in the region from point-wise
observations and one-member simulations.

## 1.   Introduction

The Southwest Pacific Ocean is a region comprising many islands, seamounts and reefs. The
interactions between the large-scale oceanic currents and the complex bathymetry lead to a complex set of oceanic
currents, transporting mass, heat and water properties, and connecting the various habitats and ecosystems across
the Coral and Tasman Seas, differently for different oceanic depths (Ceccarelli et al. 2013). As a mean, the large-
scale westward South Equatorial Current (SEC) enters into the Coral Sea where it divides into westward zonal jets
when encountering the islands of Fiji, Vanuatu Archipelago or New Caledonia (Kessler and Cravatte, 2013a;
Couvelard et al., 2008; Qu and Lindstrom, 2002). These westward zonal jets flow to the Australian coast, where
they bifurcate northward toward the Solomon Sea, feeding the Low-Latitude Western Boundary Current
(LLWBCs) and the equatorial band downstream, and southward, feeding the East Australian Current (EAC). South
of New Caledonia located around 22°S, the shallow STCC (SubTropical CounterCurrent) flows eastward above
the westward subsurface SEC. The region is characterized by a relatively high eddy kinetic energy level (Qiu and
Chen, 2004; Qiu et al., 2009); strongly sheared mean currents, vigorous western boundary currents and the
presence of islands favor the generation of instabilities and mesoscale eddies (Qiu et al., 2009; Keppler et al., 2018,
Rieck et al., 2018; Travis & Qiu, 2017).

Understanding the variability of the various currents transports in the region is of key importance to
better predict the variability of the mass, heat and freshwater transports and the water masses connections, with
implications for climate. The currents have been shown to vary at various timescales, with a dominant intraseasonal
component linked to the presence of mesoscale eddies (Qiu and Chen, 2004; Qiu et al., 2009; Cravatte et al., 2015).
At seasonal timescales, the 0-1000m large-scale transports are driven by the spin-up and spin-down of the



subtropical gyre (Kessler and Gourdeau, 2007). The interannual variability of zonal and meridional currents transports has been less documented, and is the focus of the present study.

On interannual timescales, the regional currents (and hence transports) are impacted by two main climate modes. First by the El Nino Southern Oscillation (ENSO), the dominant mode of interannual variability in the tropical band with remote oceanic and atmospheric teleconnections (see Sprintall et al., 2020 for a review). Second and farther south in the extratropics, by the Southern Annular Mode (SAM) (Thompson and Wallace, 2000). The latter is characterized by meridional shifts in westerly wind stress, potentially influencing the transports at midlatitudes. On decadal scales, wind anomalies in the southern hemisphere have also been shown to impact the sea surface height and eddy kinetic energy (EKE) (Hill et al., 2008, Cai, 2006; Holbrook et al., 2005, 2011).

Through its dipole signature in wind stress curl anomalies in the southern Pacific from the equator to about 30°S, ENSO causes large-scale sea surface height (Zilberman et al., 2013) and transport anomalies. The total 0-1000m zonal transport of waters entering into the Coral Sea between New Caledonia and Solomon Islands is reduced a few months after a La Nina, and increases a few months after an El Nino, depending on the El Nino flavor (Kessler and Cravatte, 2013b). ENSO modulates the meridional transport accordingly (e.g. Ishida et al. 2008; Lee and Fukumori 2003), and the LLWBC flowing into the Solomon Sea in the upper layer (Melet et al. 2013; Davis et al. 2012; Kessler et al. 2019).

In contrast, south of ~20°S, only a small fraction of the EAC transport variability at interannual timescales has been attributed to ENSO (Ridgway, 2007, Cetina-Heredia et al., 2014). At these latitudes, the ENSO-related wind stress curl anomalies are weak, and remote action by such curl-generated Rossby waves is hampered by the several years it takes for those to propagate from the forcing region to the western part of the basin: the ocean thus acts as a low-pass filter for ENSO induced variations (Sasaki et al., 2008). The SAM however exhibits stronger wind stress curl anomalies close to the western boundary, forcing simultaneous increases in the EAC strength and recirculation-induced meridional transports across 32°S during the SAM positive phase (Zilberman et al. 2014).

Between these two latitudes, in the 15°S-30°S band, encompassing the STCC region, the processes governing the interannual variability of the various currents are unclear. A previous attempt to explain this variability found inconsistent interannual transport variations in the region between two oceanic simulations (OFAM3 (Ocean Forecasting Australia Model) at 1/10°, and ROMS (Regional Oceanic Modeling System) at 1/12° spatial resolutions) forced by the same atmospheric forcing [Clary J, unpublished results] and observations. This suggests that the previous large-scale atmospheric interannual modes may not be the main drivers of the transports interannual variability in the region and that other mechanisms must be invoked to explain this variability.

In particular, recent studies have highlighted the importance of nonlinear internal ocean dynamics in spontaneously generating interannual variability, which is substantial in eddy active regions (e.g., Penduff et al. 2011, Sérazin et al., 2015). This so-called intrinsic variability, which can be strong, and more importantly has a random phase, can imprint large-scale and integrated quantities such as the ocean heat content (Sérazin et al., 2017; Penduff et al., 2018) and the volume transport of the overturning circulation (Grégorio et al., 2015; Leroux et al., 2017; Jamet et al., 2019). Regarding those results, the ocean appears as a chaotic system whose nonlinearities induce a sensitivity to small perturbations (Lorenz, 1963; Zanna et al., 2018) rather than an atmospherically-slaved linear system. This supports the need for ensemble ocean simulations to characterize the forced and intrinsic variabilities (e.g. Penduff et al., 2014; Bessières et al., 2017).


Mesoscale eddies, whose evolution is largely chaotic, are ubiquitous in the region south of about 20°S (Keppler et al. 2018), and the currents variability at intraseasonal timescales is much larger than at interannual timescales (Qiu and Chen 2004; Qiu et al. 2009; Cravatte et al. 2015). Sérazin et al. (2015) also show that interannual intrinsic variability is important in this region, suggesting a nonlinear upscaling from mesoscale fluctuations towards low-frequencies: nonlinear dynamics - including baroclinic instability, scale interactions and rectification (e.g., Sérazin

et al, 2018, Zanna et al., 2018) may be strong enough to eventually generate intrinsic interannual variability (Belmadani et al. 2017; Davis et al. 2014). Therefore it is plausible that part of the interannual variability arises from the intrinsic ocean variability alone.

The aim of this paper is to test this latter hypothesis, and to evaluate the respective parts of the external atmospheric variability and of the chaotic oceanic processes in driving the interannual variability of the transports in the Southwest Pacific. For that purpose, a probabilistic modeling strategy is used: an ensemble of 50 global 1/4°

ocean/sea-ice simulations, driven by the same realistic atmospheric forcing over 1960-2015 and differing only by slightly perturbed initial conditions (Penduff et al., 2014), is analyzed over 1980-2015 to interpret the currents interannual variability. This so-called "OCCIPUT" oceanic ensemble simulation along with seasonally-forced simulations have been already used to show that internal oceanic dynamics can spontaneously generate a substantial variability over a wide temporal spectrum, including interannual, decadal and long-term trends

(Gregorio et al. 2015; Leroux et al. 2018; Penduff et al. 2018; Serazin et al. 2017; Llovel et al. 2018; Penduff et al. 2019). The oceanic variability that is "deterministically" driven by the prescribed atmospheric variability (e.g. containing the ENSO or SAM atmospheric signal) is estimated from the ensemble mean evolution. Part of the prescribed atmospheric forcing may also originate from chaotic behavior in the coupled system. However as we

focus on the ocean, we will use the term "deterministic" to characterize the variability in the ocean common to all simulations. The intrinsic oceanic variability is then quantified from the random dispersion of the 50 members around the ensemble mean. To characterize this variability arising from ocean internal processes, we will use the term "chaotic" in the rest of the paper.

Section 2 describes the ensemble of simulations used, and how deterministic and chaotic variability are

quantified. It also describes the data used for validation. Section 3 quantifies the deterministic and chaotic interannual variability of the 0-1000m transports and EKE in the Southwest Pacific, with a focus on key main currents. Section 4 investigates the drivers of deterministic variability, while section 5 explores the spatial patterns of oceanic chaotic variability. We show that there are large parts of the region where interannual transport variability is firstly driven by internal chaotic ocean processes, which probably arise from a rectification of the

lower-frequency signal by mesoscale activity. This important role played by ocean-only dynamics may thus hampers our capability to identify the atmospheric drivers of the oceanic interannual variability in the Southwest Pacific and thus to predict their behavior at interannual time-scales. This has strong implications for the observing systems as discussed in section 6.

**2.    Data, model description, methods**

**2.1 Model simulations and post-processing**



The OCCIPUT ensemble is made of 50 global ocean-sea ice hindcasts, run for 56 years (1960-2015). The
model used is NEMO [Madec, 2008], implemented with a ¼° spatial resolution and 75 vertical levels (47 levels
in the first 1000m). Model parameters (numerical scheme, subgrid-scale parameterizations) are described in
[Bessières et al. 2017]. At the lateral boundaries, a free slip boundary condition is applied.

Each simulation is forced by the realistic Drakkar Forcing Set (DFS) version 5.2, based on the ERA-40 and
ERA-Interim  reanalyses (Dussin et al., 2016). After a 21-year common spinup [Leroux et al., 2018], the 50
members of the ensemble are generated in 1960 by activating a small stochastic perturbation in the equation of
state within each member [Brankart et al. 2015; Bessières et al. 2017]. This perturbation is only applied for one
year: it is switched off at the end of 1960, when the 50 members are restarted from slightly perturbed initial
conditions and driven by the same atmospheric forcing. As heat fluxes are computed using bulk formulae, they
are slightly different in the 50 members because of sea surface temperature (SST) differences. Wind stresses are
computed using absolute winds, without ocean currents feedbacks and are thus identical in each of the members.
We focus our analyses on the 1980-2015 period.

The 4D fields of temperature and velocity used in this study are available as monthly means within each
member. In addition, temperature and velocity 5-days averages are also available at specific depths for each
member. In this study, we focus mainly on two quantities: the 0-1000m integrated transports, and the EKE at the
surface and at 100m. The data processing is done as follows:
-    0-1000m integrated zonal and meridional transports are computed from monthly velocities for each member.
-    Long-term trends (very-low frequency signals) are then computed and removed from each time series within each
     member. This is done using a nonlinear, second-order local regression method (LOESS; Cleveland and Devlin
     1988), which high-pass filters time series with a 9-year cutoff period. This method preserves the length of original
time series without adverse edge effects (see Serazin et al., 2015)
-    Detrended transport time series are then low-pass filtered with a 25-months Hanning filter, in order to isolate the
     interannual variability (such filter eliminates all variability at periods lower than 1 year). Our processing thus
     confines our analyses and results on time scales between 1 year and 9 years.

We also use the 5-day zonal and meridional velocities at the surface and at 100m depth, filtered at periods
lower that 180 days with a Hanning filter, to compute the EKE. EKE monthly anomalies from the mean monthly
climatology are then computed, and the long-term trend is removed with a linear trend. Detrended time series are
then low-pass filtered with a 25-months Hanning filter, to isolate the interannual EKE modulation.

### 2.2 Estimation of the forced and intrinsic variability

For a given monthly quantity f in member i, we define for each time step t:
$$f_i'(t) = f_i(t) - f^*(t)    \qquad (1)$$
where $f^*(t) = \frac{1}{N}\sum_{i=1}^{i=N} f_i(t)$ is the ensemble mean of the N=50 members (i.e., the atmospherically forced
component). $f_i'(t)$ is the deviation from the ensemble mean (i.e. the oceanic chaotic component).
For the low-pass filtered signals F in member i, we similarly define for each time step t:
$$F_i'(t) = F_i(t) - F^*(t)    \qquad (2)$$
The intensity of the monthly forced variability is estimated with the variance of the ensemble mean time series,
the bar denoting the time average:




$$\sigma^2{}_F = \frac{1}{(T-1)} \sum_{t=1}^{t=T} \left( f^*(t) - \overline{f^*(t)} \right)^2 \quad (3)$$

The intensity of the interannual forced variability is estimated with the variance of the ensemble mean time series $F^*(t)$ low-ass filtered as explained above:

$$\sigma LF^2{}_F = \frac{1}{(T-1)} \sum_{t=1}^{t=T} \left( F^*(t) - \overline{F^*(t)} \right)^2 \quad (4)$$

The intensity of the interannual variability induced by chaotic oceanic motions is estimated by the time-mean of the ensemble variance (computed from the deviation of each member from the ensemble mean) deduced from the low-pass filtered time series $F'_i(t)$:

$$\sigma LF^2{}_I = \overline{\frac{1}{(N-1)} \sum_{t=1}^{N} \left( F'_t(t) \right)^2} \quad (5)$$

where N=50 is the number of ensemble members.

Finally, we define the deterministic variance ratio $R_{LF} = 100 * \sigma LF_F^2 / (\sigma LF_I^2 + \sigma LF_F^2)$, ie the deterministic variance in percentage of the total variance in each member (see Leroux et al. (2018), equation 8). If this ratio is greater than 50%, the deterministic variance dominates the total variance of the signal.

The interannual variances and the $R_{LF}$ ratio depend on the spatial scales considered (Serazin et al., 2015). At a model grid point, they are greater than for a field first averaged over a spatial area (see section 3). Both point-wise

and regional diagnoses will be presented; the first diagnosis for quantifying the chaotic variance and deterministic variance ratio in the framework of *in situ* data comparison, and the second one for more climate-relevant quantities.

### 2.3 Estimation of key current transports

In each member, the 0-1000m transports of key currents are computed as follows (see Figure 1, and Table 1 for a summary)

- The transport entering into the Solomon Sea is computed by integrating the 0-1000m meridional transport from coast to coast between 150°E and 162°E, along 10.5°S (labelled 1 on Figure 1a).
- The transport entering into the Coral Sea is computed by integrating the 0-1000m zonal transport along
163°E, from northern tip of New Caledonia to the Solomon Islands (labelled 2 on Figure 1a).
- The transport of the NCJ is computed by integrating the 0-1000m westward only zonal transport along 163°E, from 19°S (the northern tip of New Caledonia's reef) to 16.5°S (labelled 3 on Figure 1a); the transport of the SCJ is computed by integrating the 0-1000m westward only zonal transport along 167°E, from 22.5°S (at the southern tip of New Caledonia's reef) to 30°S (labelled 3 on Figure 1a).
- The Tasman Front transport is computed by integrating the 0-1000m eastward only zonal transport along 165°E from 40°S to 30°S (labelled 3 on Figure 1a).
- For the EAC at various latitudes (labelled 5 to 7 on Figure 1a), in addition to the southward flow only, the net transport of the southward flow and the northward adjacent recirculation is computed, following Zilberman et al. (2018): the net transport is computed by integrating the meridional 0-1000m velocity
from the coast to the offshore edge of the southward flow, and eastward to the northward EAC




recirculation edge located west of 157.5°E. The total EAC transport is also computed and shown. Intrinsic variability for both these estimates is also given.

### 2.4 Other datasets

To validate the model outputs, we use a mean absolute geostrophic velocity product, based on geostrophic shear estimated from the CARS climatological hydrographic atlas (Ridgway et al. 2002; Condie and Dunn 2006) [http://www.cmar.csiro.au/cars], and a 1000-m absolute velocity field based on Argo float drift (Kessler and Cravatte, 2013a). These 3D zonal and meridional currents are referred to as "Argo-Merged".

We also use the OSCAR (Ocean Surface Current Analysis Real- time) near-surface ocean current estimates (Bonjean and Lagerloef, 2002), from 1993 to 2015. Data are on a 1/3° grid with a 5-day resolution. The horizontal currents are directly estimated from sea surface height, surface vector wind and sea surface temperature. The OSCAR surface current is representative of the top 30m of the upper ocean.

Finally, we use the Nino3.4 index, defined as the average of the SST anomalies in the 5°S-5°N, 170°W-120°W

box, from the HadISST dataset (Rayner et al., 2003), and the PDO (Pacific Decadal Oscillation) index. The PDO is one representation of the decadal variability in the Pacific; it exhibits SST spatial patterns similar to those of ENSO in the tropics, but with a broader meridional extent (Mantua and Hare, 2002), and a time series that is primarily decadal.

### 3.    Deterministic versus chaotic oceanic transport variability

### 3.1 Assessment of the simulated ensemble mean

The ensemble-mean 0-1000m transports and surface EKE averaged over 1980-2015 are shown in Figure 1, for the numerical simulations and the observations. The model simulates realistically the main currents and zonal jets,

including the westward North and South Fiji Jets, North Vanuatu Jet (NVJ), and North Caledonian Jet (NCJ) (Couvelard et al. 2008; Gourdeau et al. 2008). The subsurface South Caledonian Jet (SCJ) (Ganachaud et al., 2008) is weaker in the simulation than in observations; the eastward flowing SubTropical CounterCurrent (STCC) flowing above in the surface layers is of similar amplitude (not shown). The Western Boundary Currents, flowing along the coast in the Gulf of Papua and then entering into the Solomon Sea (Kessler and Cravatte, 2013b; Cravatte

et al., 2011; Davis et al., 2012; Gasparin et al., 2012), and the poleward EAC and its retroflection are also correctly simulated. The eastward Tasman Front (or rather the EAC eastern extension, see Oke et al. (2019a,b)) separating from Australia between 30°S and 34°S, and meandering eastward toward the north of New Zealand (Ridgway and Dunn 2003) is suggested but not well resolved by the observations, but is clearly visible in the ensemble-mean. Also, the eastward transport east of the Solomon Islands is noisy in the observations. One missing current in the

ensemble-mean is the surface intensified eastward Coral Sea Countercurrent observed along 15°-16°S in the lee of the Vanuatu islands. As shown by Qiu et al. (2009), this current is forced by a wind stress curl dipole formed in the lee of the Vanuatu Archipelago. The absence of this dipole in the DFS forcing (not shown), probably explains this absence in the oceanic simulations.

The spread due to oceanic chaotic variability on the 36 years average transports, revealing how intrinsic

oceanic variability affects the 36 years average, is also symbolized in dots when such spread is greater than 15%



of the mean. It appears as negligible in most areas (Figure 1b). It represents nevertheless 10 to 20% of the mean current south of New Caledonia: 20% of the mean SCJ may be attributed to chaotic oceanic variability, while the EAC and Tasman Front exact latitude and meanders vary from one member to the other. This is fully consistent with the conclusions of Oke et al. (2019b) who concluded that eddies and other high-frequency features dominate
the mean flow of the so-called Tasman Front.

In terms of ensemble-mean EKE, the simulations are reasonably similar to the observations. The regions of high EKE are found at the equator, south of New Caledonia, in the STCC area, in the EAC/Tasman Front system, and in the Solomon Sea (Figure 1b). The simulated EKE is stronger than observed inside and east of the Solomon Sea, and weaker than observed in the Coral Sea west of the Vanuatu Islands, south of New Caledonia, and in the
245 EAC region. Qiu et al. (2009) suggested that the EKE in this region along 16°S is induced by the meridional shear between the eastward flowing Coral Sea Countercurrent and the adjacent westward flowing NCJ and NVJ, inducing barotropic instability. This weaker EKE may thus be linked to the absence of a CSCC in the model. In other regions, it is worth noting that the model being only eddy-permitting, a significant part of the observed EKE is missing. In fact, mesoscale activity is known to significantly increase in the NEMO model's higher resolution
configurations (e.g. Serazin et al., 2015). Finally, the different members exhibit a relatively limited dispersion in mean EKE (dots on Figure 1c, showing regions where the dispersion reaches 10 to 20% of the mean EKE), and have an amplitude similar to that of the ensemble mean EKE.

### 3.2 Variability of the 0-1000m transport

Figure 2a shows the 1980-2015 interannual variance of the 0-1000m zonal transport ensemble mean (i.e, the deterministic, atmospherically-forced, variance of these transports at interannual timescales). Figure 2b shows the intrinsic interannual variance of the 0-1000m zonal transports, and Figure 2c shows the percentage of their deterministic variance. In the ensemble mean, the interannual variability is stronger in the equatorial band, and in
the low-latitude western boundary current system, both in the Gulf of Papua and inside the Solomon Sea. In these regions, the variance is clearly deterministic, and atmospheric forcing explains more than 90% of the transports' interannual variance. The forced interannual variability is also strong in the Tasman Front, especially north of New Zealand where it reaches 400 cm$^2$.s$^{-2}$.

The intrinsic interannual variability of zonal transports is particularly strong in three regions: in the EAC and East
Auckland Current (EAUC) system regions, south of 20°S, and east and south of New Caledonia, all along the STCC path. Figure 2c reveals that in these regions, the intrinsic variability strongly dominates the deterministic variability at interannual timescales. This is particularly striking in the EAC extension region, where less than 5% of the interannual variance is deterministic. Clearly, the regions exhibiting strong intrinsic variance coincide with the regions of high EKE, suggesting that mesoscale intrinsic high-frequency variability spontaneously cascades
toward lower frequencies (Serazin et al. 2018).

The same computation is performed after a spatial smoothing of the transports over 10° in longitude and 4° in latitude, to isolate the imprint of interannual intrinsic variability at larger scales (Figure 2d). This anisotropic filtering takes into account the larger spatial scales in longitude of the zonal currents compared to the latitudinal scales. Although the relative contribution of intrinsic variability tends to be smaller for large-scale signals, the
275 region where it dominates over the deterministic variance remains unchanged. The interannual intrinsic variability



thus does not only modulate small-scale transports at model grid points, but also has a strong imprint on large-scale, climate-relevant transports.

### 3.3    Variability of key currents transports

Interannual transport anomalies of key currents in the region are also computed in each member (see section 2.1 and Table 1 for details, and Figure 1a for the locations of the sections), and their ensemble mean and spread are shown for monthly and interannual values in Figure 3. The percentage of deterministic variance for each timescale is also given, together with the maximum correlation with the Nino3.4 index. No significant correlations

are found with the SAM index. Clearly, the interannual variability of the transport entering into the Coral Sea, into the Solomon Sea, and of the NCJ are deterministic, and correlated with ENSO (see also section 4). Still, 5%, 3% and 7% respectively of these transports interannual variance is chaotic (13%, 9% and 10% for monthly means). On the other hand, the interannual variability of the transport of the SCJ, the Tasman Front and the EAC at various latitudes is largely chaotic, and strongly impacted by ocean internal processes. At 32°S, near the bifurcation

latitude, only 14% of the interannual transport variance is deterministic. In other words, inferring the interannual transport changes of these currents from a single simulation is very uncertain, as it will greatly depend on the initial conditions, and predicting them is very likely to be challenging. Surprisingly, the Tasman Front transport appears as correlated with ENSO, with a 17 months lag. Also surprisingly, at the southern end of the region at 40°S, the southern EAC extension transport is mostly deterministic, although it is known that a substantial fraction of this

transport is driven by eddies (van Sebille et al. 2012). Such a feature may arise from a substantial sensitivity of the mesoscale field itself to the interannual atmospheric forcing, which may then rectify the 0-1000m transports in a rather deterministic way. Testing such an hypothesis is left for the future.

### 3.4    Variability of EKE

The same diagnoses are performed for the interannual variability of the EKE at 100m, and are shown in Figure 4a,b,c. It is interesting to see that although a part of the EKE modulation at interannual timescales is atmospherically-forced (Figure 4a), a significant part of this interannual modulation is driven by ocean-only internal dynamics (Figure 4b,c). This is especially striking between New Caledonia and New Zealand, in the STCC/Tasman Front regions. These findings are not in agreement with Rieck et al. (2018) who concluded that the

EKE low-frequency variability was mostly deterministic in the STCC region. The possible reasons for these disagreements are discussed in Section 6. In the tropical band, the EKE interannual modulation is mostly atmospherically driven.

In addition to interannual variability, the lower-frequency evolution in EKE is also documented. Trends in both EAC and EAC extension transport have been reported in several studies (e.g. Cetenia-Heredia et al., 2014; Oke et

al., 2019), with a rapid strengthening of the eddy field since 2005 from 28°S poleward. In the ensemble mean, as in all members, a positive linear trend in EKE is indeed found from about 20°S poleward. But although the trend is largely deterministic around 25°S, its actual value within individual members is mostly random south of 30°S (Figure 4c).

### 315    3.5    Variability of the EAC separation latitude





Finally, diagnoses are also done for more integrated quantities. The latitude of the EAC bifurcation is computed in each member, following Cetina-Heredia et al. (2014) and Oke et al. (2019a). The SSH isoline contour at 28°S (where the EAC stream is well defined) associated with the maximum southward geostrophic current is identified, and followed southward until it veers eastward. The corresponding latitude is identified as the separation latitude
(Figure 5a), and its ensemble mean and spread are shown in Figure 5b for monthly and interannual time series. The latitude where the EAC separates from the coast can vary abruptly at intraseasonal timescales, as eddies detachments lead to northward displacements of the separation point. This latitude is strongly varying from one member to the other, as eddies development and detachments are not synchronous. Interestingly, the interannual variability of the EAC separation latitude is also strongly dominated by intrinsic variability, since the deterministic
variance only represents 34% of the variance. No significant correlation with ENSO is found, whereas Cetina-Heredia et al. (2014) suggested that the relaxation of an ENSO event induces a shift in the EAC separation latitude. It confirms that the definition of a bifurcation latitude, and of a well-defined continuous eastward flow emanating from the coast is not an adequate description of the eddying circulation (Oke et al., 2019b).

## 4.  Drivers of deterministic variability

### 4.1      0-1000 interannual transport

The extent to which the interannual variability of the transports in the southwestern Pacific is driven by ENSO
or other climate-related indices (contained in the prescribed atmospheric forcing) is now explored. Firstly, it is explored within one member picked randomly. Figure 6a shows the maximum lagged correlation between the Nino3.4 Index and the 0-1000m transports in member 39, with the corresponding lag in months (Figure 6c). Positives correlations mean that the transport increases in relation to Nino3.4, and negative lags that Nino3.4 lead the transport variations. Significant correlations are found north of 20°S, with positive anomalies in the equatorial
band and south of 10°S a few months after Nino3.4. This corresponds to eastward anomalies at the equator, and westward anomalies in the transport entering into the Coral Sea. Positive (northward) anomalies in the LLWBC through the Solomon Sea are also found, in agreement with previous findings. A few months after an El Nino event, a dipole of wind stress curl anomalies is found over the South Pacific, from 5°S to 30°S, and 150°E to 140°W. These wind stress curl anomalies induce an increase in the westward SEC transport entering into the Coral
Sea (Kessler and Cravatte, 2013), and an increase of the NGCU into the Solomon Sea (Melet et al., 2013; Davis et al., 2012; Kessler et al., 2019) mainly in the top 250 m. The lag increases with latitude, as expected from the poleward decrease in Rossby waves westward phase speed, as these waves are the means by which the curl modifications are transmitted across the ocean toward the western Pacific. It is also found that the EAC poleward velocity increases 6 to 15 months after an El Nino, north of 30°S. Further south, the map is noisy, and no significant
correlations are found at any lag.

The same analysis is now done with the ensemble mean. Interestingly, in addition to the significant correlations found in member 39, additional patterns emerge. Some of the transports south of New Caledonia also exhibit a correlation with ENSO. The SCJ (around 27°S, 170°E) decreases a few months after an El Nino, and the latitude of the Tasman Front maximum transport also emerges as significantly correlated with ENSO, shifting northward
15 months after an El Nino. This reveals that the chaotic oceanic variability can mask the ENSO imprint on





interannual transport variability south of 20°S, and that an ensemble mean allows to better identify this ENSO imprint. Whether an ensemble mean computed from more than 50 members would yield more regions with significant correlations is not known, and the number of members needed to converge would be worth investigating.

Similar correlations were computed with the PDO index and the SAM index (not shown). The EAC separation latitude and the Tasman Front latitudinal extension appear to move southward 10 to 15 months after a positive SAM. During a positive PDO phase, the patterns of transport changes are very similar to the ENSO ones, with a stronger signal in the Tasman Front region, where the transport increases while the EAC southward extension decreases. These decadal anticorrelated variations in both currents are consistent with previous findings (Hill et

al. 2011; Sasaki et al. 2008). All these correlations are somewhat clearer in the ensemble mean than in individual members.

### 4.2    EKE

Figure 7a shows the maximum lagged correlation between the Nino3.4 Index and the 100m EKE from the ensemble mean, with the corresponding lag in months. Two regions of significant correlation emerge: the tropical band, including the Solomon Sea, and the Tasman Front region. A few months after an El Nino, EKE decreases in the tropics, consistently with findings from Tchilibou et al. (2020). 15 months after an El Nino, EKE conversely increases in the Tasman Front area, when the Tasman Front is stronger and shifted northward.

The EKE is also modulated at decadal timescales, in connection with the PDO in the STCC region, as suggested previously by Rieck et al. (2018) and Travis and Qiu (2017) (Figure 7b). During positive phases of the PDO, the EKE decreases in the whole STCC band, in addition to the equatorial band. The processes explaining this decadal deterministic modulation have been investigated in Rieck et al. (2018) and Travis and Qiu (2017). EKE is modulated through a mix of processes including changes in large-scale zonal currents strength, associated vertical

shear and stratification. These changes, forced by decadal wind changes in the subtropical gyre, work simultaneously to modify the baroclinic instability intensity, and the EKE levels. As shown in Figure 4c, this deterministic forcing however only represents 10 to 30% of the EKE variability at interannual timescales. It is worth noting that these correlations do not clearly emerge in individual members (see dots in Figure 7).

We further examine the correlation between the vertical shear (defined as the difference between the zonal current

at 600m and at 200m) (see Travis and Qiu, 2017), and the EKE levels in the STCC region (160°E-180°, 22°S-28°S) (Figure 8). In the ensemble mean, a significant correlation (0.75) is found between the shear and the EKE (Figure 8c, red dashed line), as also found in altimetric data and a reanalysis in Travis and Qiu (2017) or an OGCM simulation in Rieck et al. (2018). In each individual member however, there is not always a significant correlation between vertical shear and EKE amplitude. The vertical shear appears to be driven at least partly by the

atmospheric forcing, as the members do not exhibit a substantial dispersion (Figure 8a), and 80% of the variance is deterministic. On the contrary, the EKE interannual modulation is dominantly intrinsic (Figure 8b), as 40% of the variance only is deterministic. These results highlight the complexity of EKE generation, and the interplay of different forcing processes, whose relative importance varies. The vertical shear, the stratification, and the propagations of eddies from remote regions may all influence the EKE levels. Travis and Qiu (2017) showed that

these processes vary in time, and can alternately work to attenuate or reinforce each other. Here, we suggest in





addition that they vary differently in the members, and can alternately attenuate or reinforce each other in the different members.

### 5. Spatio-temporal structure of the chaotic oceanic variability

What are the processes at work producing the interannual intrinsic variability? How do mesoscale eddies imprint the transport interannual variability? Is the chaotic oceanic variability spatially coherent at large scale? To provide some possible hints of answers, we now look at individual members, and analyze the chaotic variability in these members with EOF (Empirical Orthogonal Functions) analyses, both for monthly and interannual time series.

Two regions of large oceanic chaotic variance are examined in more detail: a- the EAC extension region, from
30°S to 40°S, and b- the STCC area, where both transport and EKE modulation are dominantly chaotic. In each member, the deviations from the ensemble mean of zonal and meridional transports at monthly and interannual timescales are first computed. A multi-variable EOF analysis is first performed on both components of the transport (Dawson, 2016). Additionally, the 50 members are combined to produce a 50x(1980-2015) time series, on which a multi-variable EOF analysis is performed, on zonal and meridional transports. This allows isolating
the spatial patterns of chaotic variability common to all members, while allowing them to have different timings revealed by their PC (Principal Components). The first 2 EOFs emerging from the individual EOF analysis are similar to those emerging in the 50-members combined EOF analysis, giving confidence in the robustness of the signals.

In the EAC region (Figure 9), the monthly EOFs reveal the presence of mesoscale eddies, with periods of 100-180
415   days, as shown by previous studies (e.g. Oke et al., 2019). Figure 9c shows the associated monthly and interannual PCs extracted for member 1, and Figure 9d the other PCs for the 50 members. The eddies amplitude envelope is modulated in time. In all members, the characteristics of the eddies are similar (periods, amplitude), as expected since they share the same model dynamics. However, their phasing differs, and their interannual modulation phase also differs from one member to the other. Further investigating the non-linear interactions leading to the transport
interannual variability, probably arising from eddy low-frequency rectification, would be interesting but is beyond the scope of this study.

In the STCC region, the spatial structure of the chaotic variability is quite different (Figure 10 a, b). In terms of zonal transport, it consists in zonally elongated structures, alternating meridionally, slanted in a southwest-northeast direction, and slowly propagating southward, as can be seen when comparing the two first EOFs and
PCs. Such striations-like structures have already been noticed in the western half of the subtropical basins with low-passed altimetric observations (Maximenko et al., 2005) and on global maps of sea level intrinsic variability (Serazin et al., 2015, their Figure 4). These fluctuating striations might be generated by an incomplete anisotropic inverse cascade from small-scale intraseasonal structures like eddies towards low-frequency zonally-elongated structures (see section 6 of Serazin et al., 2018 for a complete discussion). Because some parameters might prevent
the inverse cascade to reach the basin scale - including the presence of a meridional mean flow (Chen and Flierl, 2015), the intensity of bottom friction (Berloff et al., 2009; 2011) and the presence of meridional boundaries (Lacasce, 2002) - these striations cannot develop into persistent zonal jets as in the atmosphere but rather behaves as weak latent jets that shows up only on filtered signals such as the EOF decomposition performed here.

Interestingly, monthly and interannual EOFs and PCs are similar (Figure 10c), revealing that large-scale intrinsic
variability in this area is dominated by a low frequency (3-4 year timescale) signal. These anomalies are not in



phase in the different members, confirming that they arise from ocean-driven intrinsic processes, potentially through nonlinear scale interactions (i.e., an inverse cascade) as discussed above.

**6.  Discussion and conclusion**

In this study, the origin and features of interannual variability of the transports and EKE in the Southwest Pacific have been investigated using a probabilistic modeling strategy. A large ensemble of 50 simulations performed with the same OGCM at eddy permitting resolution, perturbed initially but driven by the same realistic atmospheric forcing over 1980-2015, has been analyzed to disentangle the atmospherically-forced, deterministic

ocean variability and the internal, chaotic ocean variability. This approach had previously been used to show that chaotic oceanic processes can have more impact than the deterministic atmospheric variability on the low-frequency variability and trends of regional transport, sea level or oceanic heat content, particularly at mid-latitudes (Leroux et al. 2018; Llovel et al. 2018; Serazin et al. 2017).

Our results show that in the Southwest Pacific, the interannual variability of the transports is strongly

dominated by chaotic oceanic variability south of 20°S. In the tropics, while interannual variability of transports and EKE modulation is largely deterministic and explained by ENSO, oceanic chaotic processes may still explain 10 to 20% of the interannual variance at large-scale and can reach 40 to 60% punctually in areas of complex topography such as the Gulf of Papua, inside the Solomon Sea, and east of the Solomon Islands. Regions of strong chaotic variance generally coincide with regions of high mesoscale activity; the interannual chaotic ocean

variability is greater when the EKE is larger, suggesting that a spontaneous inverse cascade is at work from mesoscale toward lower frequencies and larger scales in agreement with Serazin et al. (2018).

The spatial-temporal features of the oceanic chaotic variability are complex, and varies from region to region. Interestingly, as the simulations share the same dynamics, they produce chaotic structures with similar spatial characteristics but with different phasing. In the STCC, the low-frequency imprint of these structures is

zonally elongated, and its spatial organization is likely the result of an incomplete anisotropic inverse cascade due to nonlinear scale interactions (e.g., Sérazin et al., 2018). In other regions, they are eddy-shaped. The physical processes governing this chaotic variability remain to be fully understood.

Our results on EKE contrast with the findings of Rieck et al. (2018), who concluded that the EKE decadal variability in the STCC region was, uniquely among the subtropical gyres, mostly deterministic, driven by wind-

forced decadal changes. In agreement with Travis and Qiu (2017) analyses of observations, they explained these decadal EKE modulation by decadal changes in vertical shear, driven by wind stress curl changes in the South Pacific, while still recognizing that other processes such as changes in stratification or density anomalies remotely forced also contribute to modulate the EKE low-frequency changes. The disagreement between our and these authors conclusions is likely due to our different modeling strategy, which gives access to both intrinsic and

atmospherically driven oceanic fluctuations: Rieck et al. (2018) used a long climatological run to study the sole imprint of intrinsic interannual variability. Here, we use 50 members with fully varying atmospheric forcing, and we found a link between the decadal variations of vertical shear in the STCC region and EKE modulation in the ensemble mean. However, in each member, we found that other processes (probably such as stratification changes, and propagations of eddies from remote regions) all contribute to modulate the EKE levels, the contribution of the

vertical shear having a different relative importance in the members. These results highlight the complexity of



EKE generation, and the interplay of different forcing processes; they also emphasize the importance of taking simultaneously into account the chaotic ocean behavior and the imprint of the atmospheric variability in an ensemble approach. It is also worth noting that our results do not question the existence of deterministic, forced interannual variability of EKE in the area; our results mostly emphasize that the chaotic ocean interannual

variability may overwhelm this deterministic variability over large regions.

The fraction of interannual variability explained by chaotic oceanic processes depends on the spatial scale considered. At model grid points, this fraction is greater than for spatially integrated variables, or even for large-scale transports across sections. Yet, our results show that for some key currents, section-integrated transport variability is dominantly chaotic. Moreover, our study reveals that chaotic variability may result in large-scale,

spatially coherent signals at interannual timescales. The EAC separation latitude variability (defined where eddies detach off the EAC) at interannual timescales is also largely chaotic.

These results have important implications for the understanding and forecasting of the transports in the region, and for observational strategies. Firstly, they show that the impact of climate modes of variability such as ENSO or PDO on the interannual variability of transports and EKE may be hidden by oceanic internal variability,

and that an ensemble of simulations is needed to bring out this signature. This clearly hampers our capacity to attribute the forced origin of the interannual variability of transports from a single simulation.

Importantly, these results highlight that interpretation of *in situ* or satellite observations in areas of high intrinsic variance should be done with caution. While the low-frequency signals amplitude does not strongly vary between members, their phase may be highly random because of chaotic oceanic processes, and may be largely

uncorrelated with the atmospheric variability. Figure 11 illustrates how chaotic ocean variability can imprint synthetic (i.e. model-simulated) mooring measurements at various time scales. The histograms of the meridional velocity at 100 m for the 50 members at existing moorings locations in the EAC (at 27°S), and in the Solomon Sea Strait (Alberty et al. 2019; Sloyan et al. 2016) are shown at a given date, for 5-day averages, monthly averages, yearly averages and 2005-2015 averages. It is shown that even for an instrument deployed in a mostly deterministic

current such as the NICU in the Solomon Strait, the dispersion of values can be large. If one considers that an *in situ* observation samples one ocean state randomly picked from an ensemble of possibilities, the larger the intrinsic variability, the less representative an observation is, and the less its interannual variability can be attributed to the atmosphere, and thus the less it can be predicted and understood deterministically. This also suggests that assimilation of these data in numerical simulations should consider this additional source of uncertainty. The

repercussion on errors in gridded products based on individual profiles, such as in Argo temperature and salinity gridded products, or satellite tracks gridded products should be usefully evaluated (also see in Penduff et al. (2018) their Figure 2 and the associated discussion about the representativeness of temperature measurements). Such studies are currently underway.

It should be kept in mind that we used an eddy-permitting (1/4°) resolution that sits at the limit of presently

affordable long-term global ocean large ensembles, but in which only part of the mesoscale activity is simulated. It is probable that at higher resolution, with the increased mesoscale activity, inverse cascades of energy toward large scales and low frequencies are stronger: this has been shown in terms of sea level by Serazin et al. (2015), who suggested a direct feeding of low-frequency chaotic variability by mesoscale activity both within and away from eddy-active regions. Gregorio et al (2015), however, have suggested that the forced variability of the AMOC

is also likely to increase when switching from 1/4° to 1/12° resolution; whether this also holds for the variables



investigated here needs to be verified, but these authors did not report a substantial sensitivity of forced-to-total variability ratios (noted RLF here) to this resolution increase. More generally, it would be interesting to assess whether the OCCIPUT ensemble gives a robust estimate of the ocean chaotic variability; comparisons with other ensemble simulations (such as that performed at 1/10° by Nonaka et al. (2020) would be of use for such an assessment.

We also used forced and not coupled oceanic simulations, which is an indispensable step to isolate the chaotic ocean-only contributions. The importance of air-sea interactions at mesoscale are however known to be significant. The wind stress variability is impacted by the ocean current feedback, which could damp the EKE eddy-scale variability in return (e.g. Renault et al. 2017, 2019). Intrinsic variability is also likely to impact the air-sea heat flux variability through sea surface temperature fluctuations and affect the atmospheric variability in return. These possible effects should be investigated in the future as the observed variability reflects the fully coupled climate system.



*Data availability:* The model dataset used for this study is available on request (thierry.Penduff@cnrs.fr).

*Author contributions:* Thierry Penduff lead the OCCIPUT project and run the experiments. Sophie Cravatte performed the analyses, prepared the manuscript, with contributions from all co-authors. All authors have contributed to the interpretations of the results and to a significant part to the presented scientific work.

*Competing interests:* The authors declare that they have no conflict of interest.

*Acknowledgements:* The results of this research have been achieved using the PRACE Research Infrastructure resource CURIE based in France at TGCC. This work is a contribution to the OCCIPUT and PIRATE projects. OCCIPUT has been funded by ANR through contract ANR-13-BS06-0007-01. PIRATE is funded by CNES through the Ocean Surface Topography Science Team (OST-ST). The authors would like to thank L. Bessières and W. Llovel for helping with extracting the files from the OCCIPUT simulations. The authors wish to acknowledge Ssalto/Duacs AVISO who produced the altimeter products, with support from CNES (http://www.aviso.altimetry.fr/duacs/), the OSCAR Project Office, and the CARS compilation of hydrographic data (http://www.cmar/csiro.au/ cars). The authors finally wish to acknowledge the use of the Ferret program for analysis and graphics in this paper. Ferret is a product of NOAA's Pacific Marine Environmental Laboratory. (Information is available at http://ferret. pmel.noaa.gov/Ferret.).

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


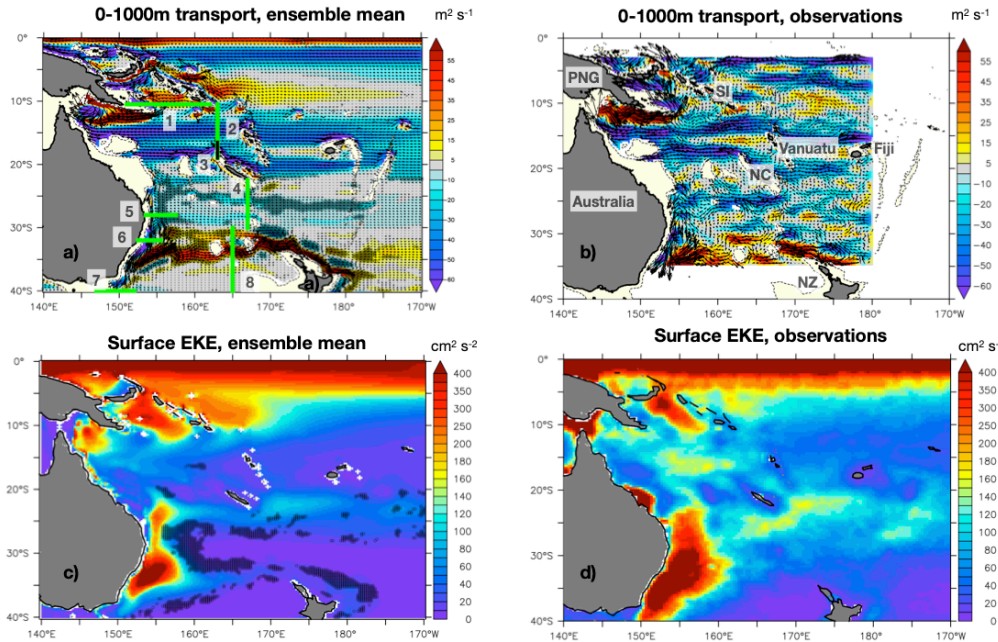

Figure 1: a) 1980-2015 time averaged 0-1000m transport from the ensemble mean. Colors show zonal transport (positive values mean eastward, negative values westward), and arrows total transport. The hatched areas show regions where the spread of these 1980-2015 time averaged 0-1000m transport are at least 15% of the ensemble mean. The green thick lines (and the black thick line) show the main current sections shown in Figure 3, with the associated numbers corresponding to the caption of Figure 3. b) 0-1000m transport from the Argo-Merged observations; colors show zonal transport. The names of the main islands are also indicated. NC: New Caledonia; NZ: New Zealand; SI: Solomon Islands and PNG: Papua New Guinea. c) Time averaged ensemble mean of the 1980-2015 surface EKE. The hatched areas show regions where the spread of these 1980-2015 time EKE are between 10% and 20% of the ensemble mean d) 1993-2015 time averaged surface EKE from OSCAR product.

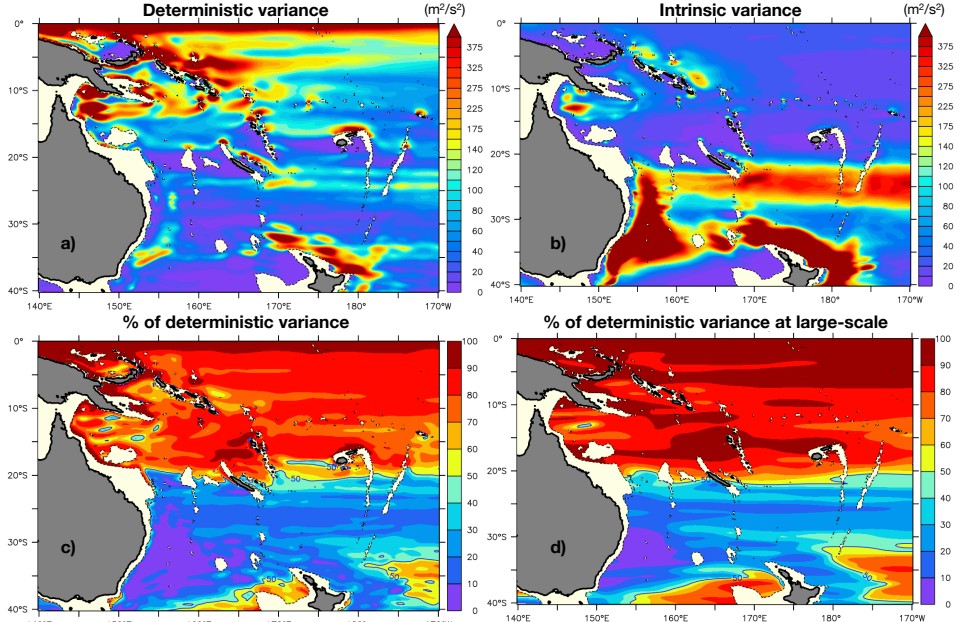

Figure 2: a) Deterministic interannual variance: 1980-2015 variance of ensemble mean low-pass filtered 0-1000m zonal transport b) Intrinsic interannual variance: ensemble mean of the squared deviation of the low-pass filtered 0-1000m zonal transport in each member from the ensemble mean. c) Ratio $R_{LF}$ of the deterministic variance on the total variance, given in percentage of the deterministic variance. d) Same as c, but for transports first spatially smoothed in each member.


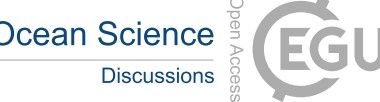
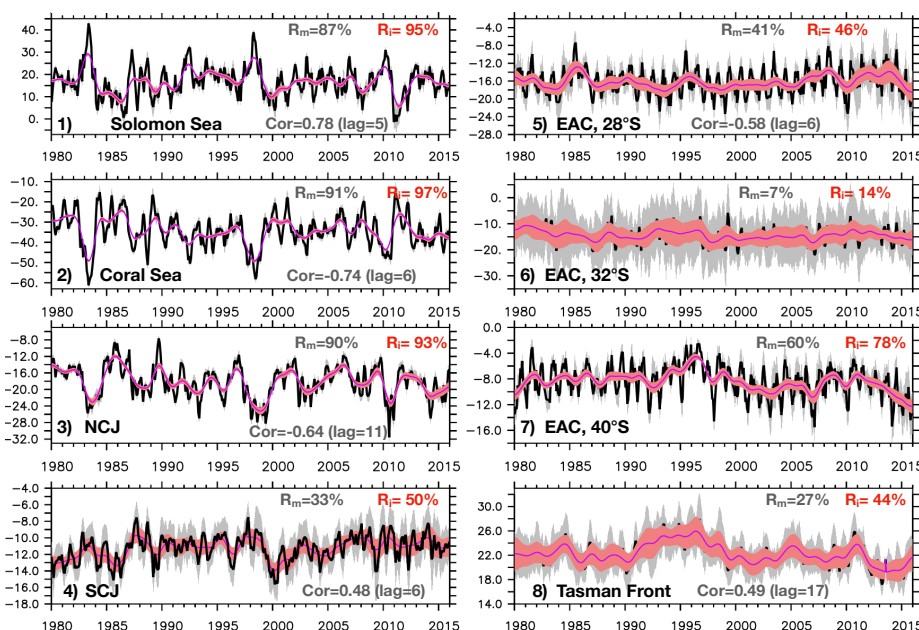

Figure 3: Time series of the 0-1000m integrated transports across various sections (in Sv). In black: monthly values from the ensemble mean; in grey shades, ensemble mean plus or minus the monthly ensemble spread. In red: interannual values from the ensemble mean, and the red shades, ensemble mean plus or minus the interannual ensemble spread. The percentage of the deterministic variance is given as Rm for the monthly values and Ri for the interannual values. The maximum correlation of the transport with the Nino3.4 index is also given, with the associated lag in months. 1) meridional transport through the entrance of the Solomon Sea (positive is northward) 2) zonal transport through the entrance of the Coral Sea (positive is eastward) 3) westward transport of the NCJ 4) westward transport of the SCJ 5) meridional total transport of the EAC at 28°S (see section 2.3 for definition). Rm and Ri for the southward only meridional transport are 44% and 58%, respectively. Rm and Ri for the northward recirculation branch are 21% and 50%, respectively. 6) meridional total transport of the EAC at 32°S. Rm and Ri for the southward only meridional transport are 10% and 29%, respectively. Rm and Ri for the northward recirculation branch are 6% and 22%, respectively. 7) meridional transport of the EAC at 40°S. 8) Zonal transport of the Tasman front. See text for details.



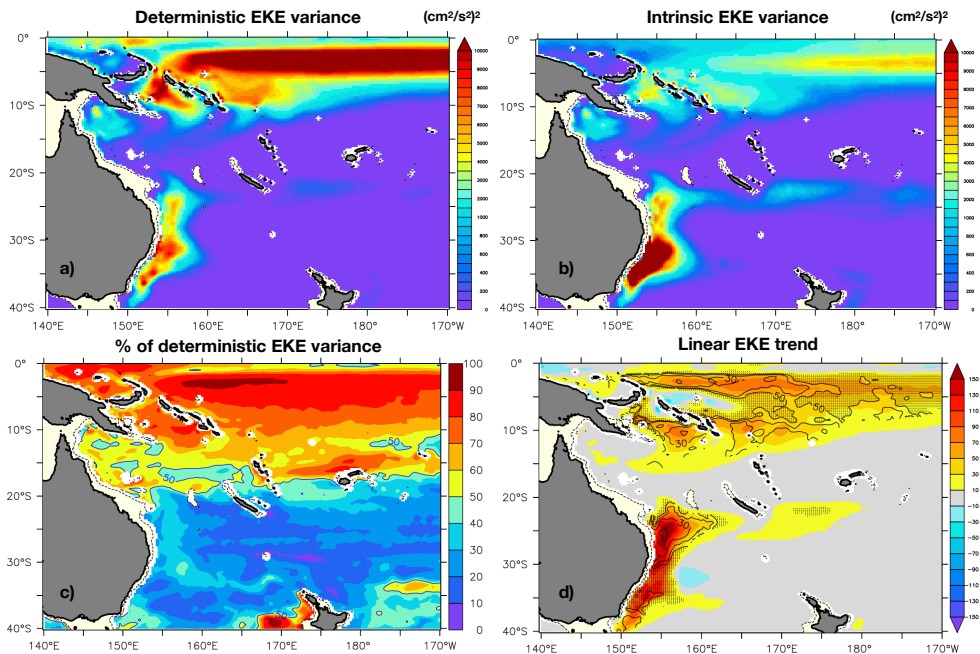

Figure 4: a) Deterministic interannual variance of the EKE: 1980-2015 variance of ensemble mean low-pass filtered 100m-EKE b) Intrinsic interannual variance: ensemble mean of the squared deviation of the low-pass filtered surface EKE in each member from the ensemble mean. c) Ratio $R_{LF}$ of the deterministic EKE variance on the total EKE variance, given in percentage of the deterministic EKE variance. d) 100m-EKE ensemble mean linear trend (in colors), and ratio of the spread between these linear trends and the deterministic trend (contours in percentage). The hatched areas show regions where the spread of these 1980-2015 EKE trends is higher than 50% of the ensemble-mean trend.





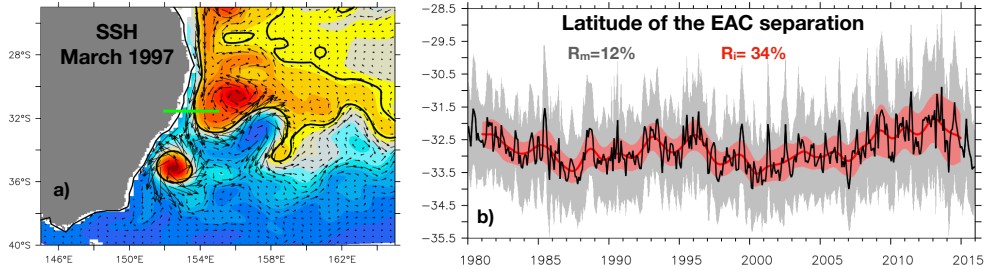

Figure 5: a) Sea Surface Height (SSH) snapshot from one member on 15 March 1997 (colors), with the contour of the isoline associated with the maximum southward geostrophic current at 28°S (thick black line), and geostrophic surface currents (vectors). The latitude of the EAC separation on that snapshot, at which the isoline veers eastward, is indicated in thick green line. b) Time series of the latitude of EAC separation. In back: monthly values from the ensemble mean; in grey shades, ensemble mean plus or minus the monthly ensemble spread. In red: interannual values from the ensemble mean, and the red shades, ensemble mean plus or minus the interannual ensemble spread. The percentage of the deterministic variance is given as Rm for the monthly values and Ri for the interannual values.




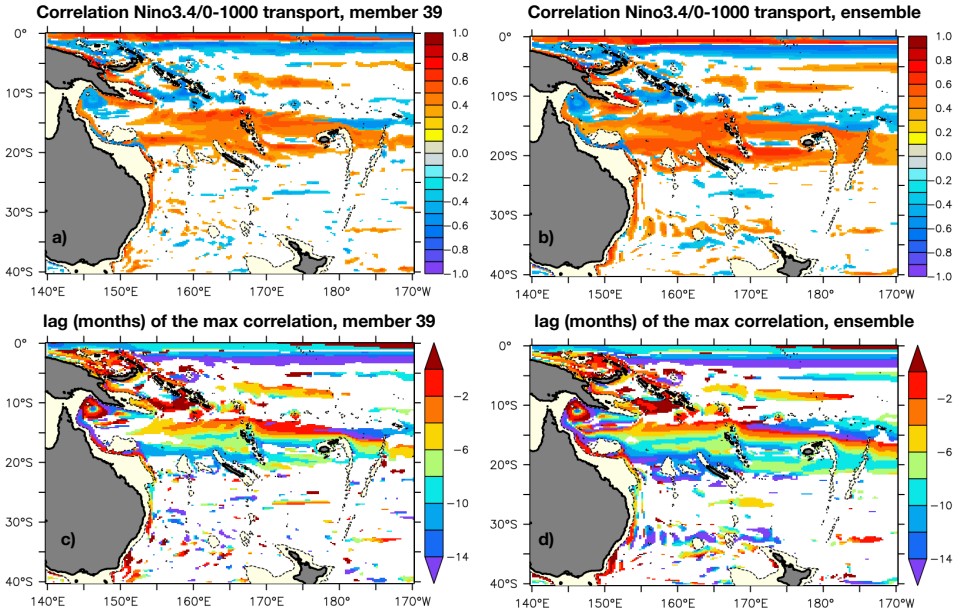

Figure 6: Maximum lagged correlation (a, b) and corresponding lag in months (taken between -15 and 5 months) (c, d) between Nino3.4 index and the 0-1000m interannual transport variations for (a, c): member 39 of the ensemble, and (b, d) the ensemble mean. A negative lag means that Nino3.4 lead the transports variations. Only significant correlations are plotted.





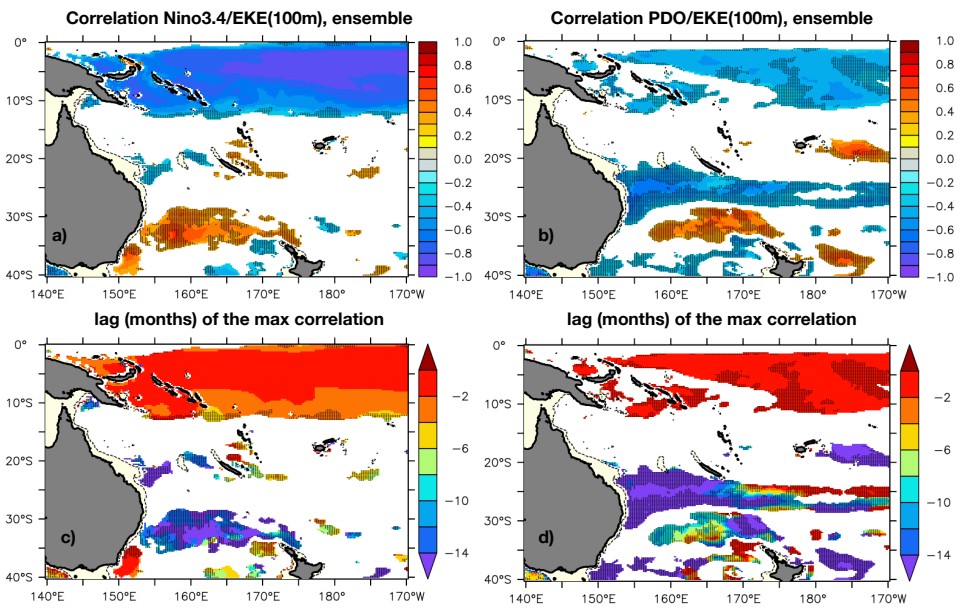

Figure 7: Maximum lagged correlation (a) and corresponding lag in months (taken between -18 and 0 months) (c) between Nino3.4 index and the 100m-EKE interannual variations for the ensemble. Only significant correlations are plotted. A negative lag means that Nino3.4 lead the EKE variations. (b, d): same as (a, c) for the PDO index. In all panels, the hatched areas show regions where the same maximum lagged correlation for 100m-EKE interannual variations in one member picked randomly (member 39) are not significant.

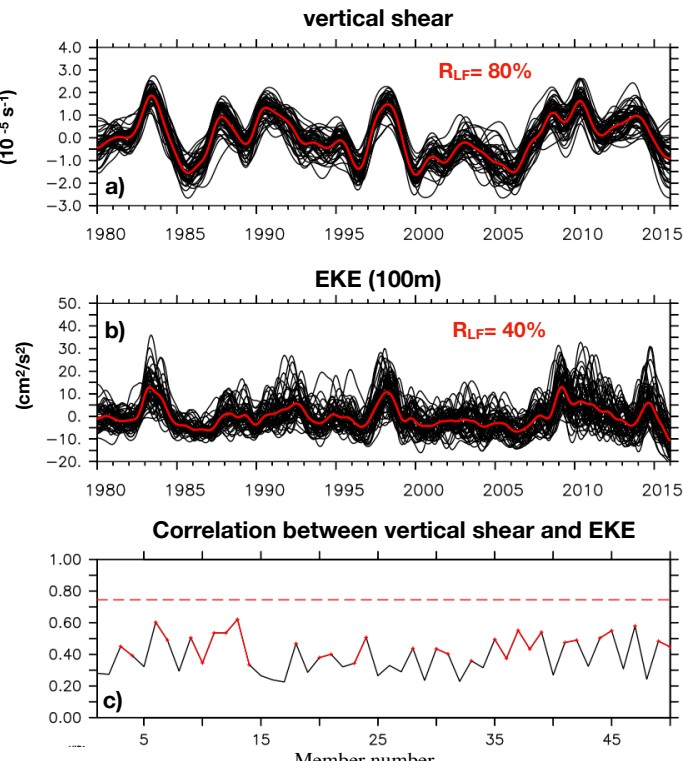

Figure 8: Time series of the vertical shear ($10^{-5}$ s$^{-1}$) (a), EKE at 100m (cm$^2$ s$^{-2}$) (b) interannual anomalies averaged in the STCC region (28°S-22°S, 160°E-180°E). In black: values from the 50 members, in red, from the ensemble mean. (c): Correlation between the vertical shear and the EKE interannual anomalies, for all the members (in black), the x-axis corresponding to the 50 members, and for the ensemble mean (in dashed red). The correlations significantly different from zero at the 95% confidence level are overlaid in red. The ratio $R_{LF}$ of the deterministic variance on the total variance, given in percentage of the deterministic variance, are given for the shear and the EKE.




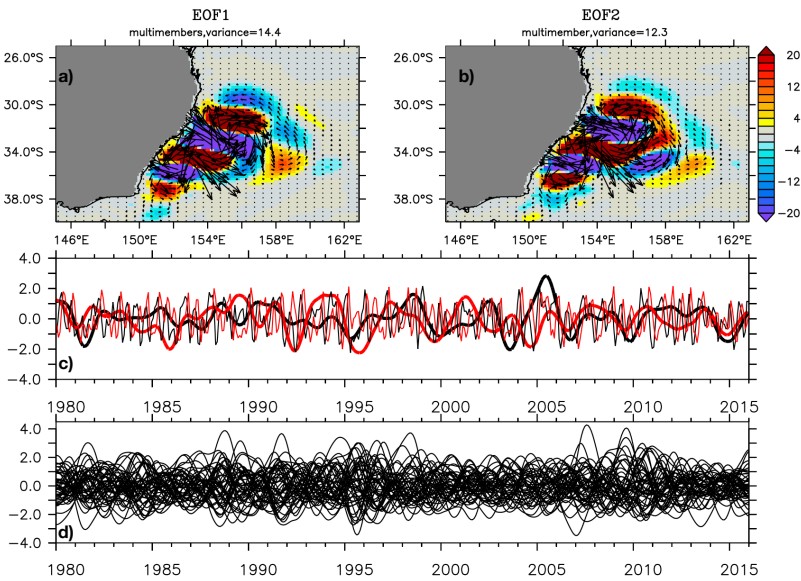

Figure 9: First (a) and second (b) spatial EOFs in the EAC region (25°S-40°S, 145°E-163°E) of the 50 members deviations from the ensemble mean of zonal and meridional transports, considered as a single time series. (c) PCs of the first (black) and second (red) mode, from monthly chaotic transports in thin lines, and interannual chaotic transports in thick line. (d) The 50 first PCs from the 50 chaotic members interannual transports.




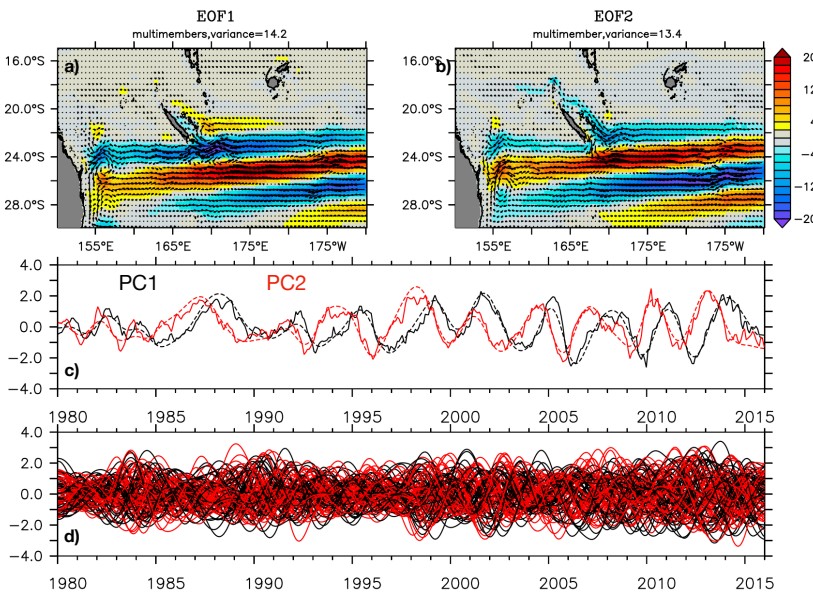

Figure 10: First (a) and second (b) spatial EOFs in the STCC region (15°S-30°S, 150°E-170°W) of the 50 members deviations from the ensemble mean of zonal and meridional transports, considered as a single time series. (c) PCs of the first (black) and second (red) mode, from monthly chaotic transports in thin lines, and interannual chaotic transports in dashed line. (d) The 50 first (black) and second (red) PCs from the 50 chaotic members interannual transports.





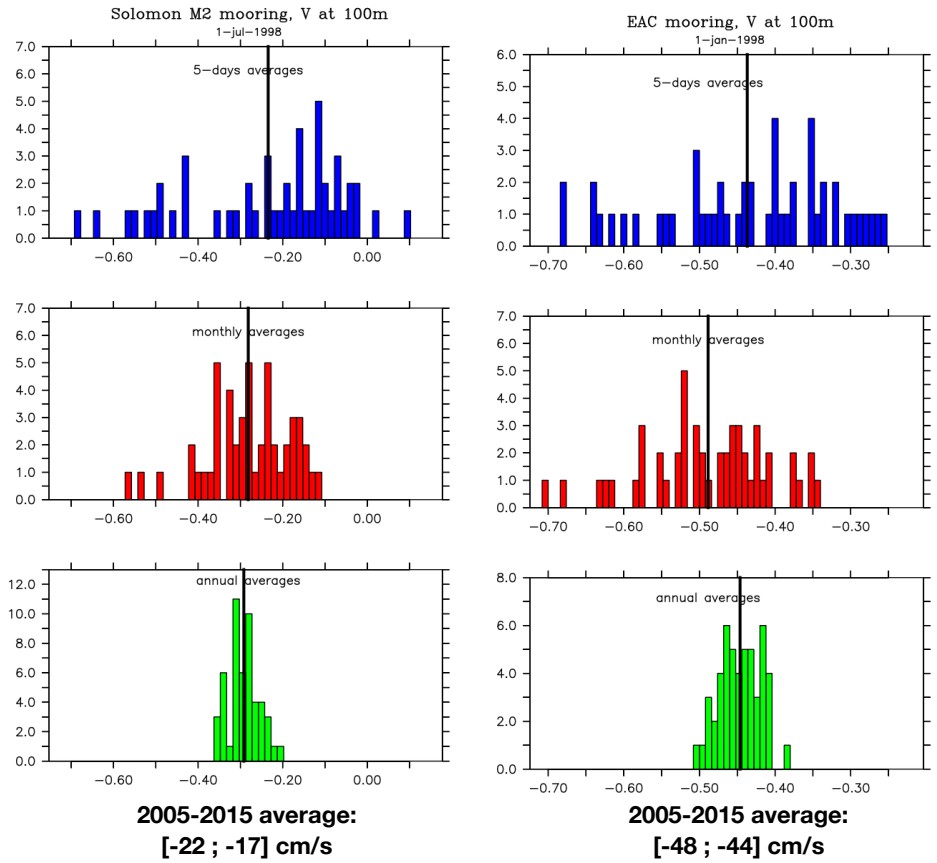

Figure 11: Histograms of velocity among the 50 members, for the meridional velocity at 100m from 5-days average values in blue (a,d), monthly averages in red (b,e) and annual averages in green (c,f), at two locations around a particular date (on 1 July 1998 inside the Solomon Strait, at 154.2°E, 5.36°S (a,b,c) and on 1 January 1998 in the EAC, at 154°E, 27.06°S (d,e,f)). The value of the ensemble mean is also plotted in black lines.





| Number | Transport | Latitudes | Longitudes | Sign of the current |
|---|---|---|---|---|
| 1 | Solomon Sea entrance | Along 10.5°S | 150°E-162°E | Total transport |
| 2 | Coral Sea entrance | 19°S-10.5°S | Along 163°E | Total transport |
| 3 | NCJ | 19°S-16.5°S | Along 163°E | Westward only |
| 4 | SCJ | 30°S-22°S | Along 167°E | Westward only |
| 5 | Tasman Front | 40°S-30°S | Along 165°E | Eastward only |
| 6, 7 and 8 | EAC at various latitudes (including the EAC southward extension) | 28°S, 32°S, 40°S | From the coast to 157.5°E | Total, southward only, norward only |

Table 1: Computation of the 0-1000m transport of key currents shown in Figure 3. The numbers correspond to the sections shown in Figure 1.