# Peer review of "Imprint of chaotic ocean variability on transports in the Southwest Pacific at interannual timescales"

_Ocean Science, 2020_

## Referee Comment (RC1) · Jan Klaus Rieck (Referee) · 11 Nov 2020

**General comments**

This study investigates the transport variability in the Southwest Pacific on interannual time scales with the help of the 1/4°, 50-member ensemble simulation OCCIPUT. The focus is put on the influence of intrinsic (chaotic) oceanic variability on the observed transports in contrast to the deterministic variability forced by the atmosphere. The study also points out a general need for ensemble simulations to better quantify oceanic variability on interannual time scales, and includes a valuable discussion of the study's implications for the interpretation of e.g. observations.

[Figure]

One important scientific value of this study is the validation or falsification of some results that have already been published previously, but were based on single-member simulations or point-observations. If these results can (cannot) be reproduced with the OCCIPUT ensemble, they are much more (less) significant, due to the smaller errors using an ensemble and the advantage of estimating the chaotic variability that emerges internally in the ocean.

The study is well-structured and comprehensive, and monstly needs some clarifications and minor technical revisions as listed below in the "Specific comments" and "Technical Comments". However, there are three issues, that require some attention:

1. In the introduction, the latitude band 15-30°S is identified as a region where many processes are not well understood and the OCCIPUT ensemble can help clarify. However, the current systems outside this latitude band (where previous studies reached more consistent conclusions) also recieve substantial attention in this mansucript. I therefore recommend, to more clearly point out that the validation of previous results with this novel modeling approach is an important aspect of this study, and not only the investigation of processes that could not be explained satisfacatorily so far.

2. Another issue that needs some more work is the discussion about the results of this study in contrast to Travis and Qiu (2017) and Rieck et al. (2018). While I have no doubt that the general conclusion you draw is valid and the interannual variability in the STCC region is mostly intrinsic, there are two points that should be clarified:

- The different regions used in this study compared to Travis and Qiu (2017) and Rieck et al. (2018). Given that the simulations are global, you could easily extend the region you use to define the STCC and thus make it better comparable to the regions used in the other two studies. (Also see specific comment ll. 463-465 below)

- While Travis and Qiu (2017) and Rieck et al. (2018) investigate decadal variability, this study focusses on interannual variability. This, and the implications, should be made more clear.

3. Additionally, the manuscript would benefit from an overview map. I suggest adding a new, large Fig. 1 to the manuscript, showing the bathymetry of the Southwest Pacific and the locations of the different seas (e.g. Coral Sea, Solomon Sea, etc ...), currents, and islands.

**Specific comments**

*1. Introduction*

*l. 30 "equatorial band":* Do you mean the current bands?
*l. 38 "These currents":* Please specify which currents are referred to here; is the statement valid for all currents mentioned in the paragraph above?
*l. 48 "wind anomalies in the southern hemisphere":* Are these wind anomalies related to the SAM as well?
*l. 57 "[...] south of 20°S [...]:* In the paragraph above, ENSO is said to affect wind stress curl to 30°S. Here however, ENSO's influence is said to be restricted to North of 20°S. This is inconsistent.
*l. 64 "Between these two latitudes [...]:* Does this refer to 15-30°S? If yes, there is no need for this first part of the sentence.

*2. Data, model description, methods*

*l. 130:* I suggest to include the discussion about the impact of using a coupled system here (ll. 521-527). It is an important discussion but it does not fit at the end of the manuscript in my opinion.
*l. 131:* Is there any specific reason to restrict the analyses to the period 1980-2015? Does the ensemble need the 20 years for the solutions to sufficiently diverge?
*l. 131:* How is the PDO index defined?
*l. 144-145:* Do you use the 180-day low-pass filtered velocities as the mean in the EKE

calculations? Or as the deviations from that mean? This is not clear.

**3. Deterministic versus chaotic oceanic transport variability**

*I. 219:* I suggest to better specifiy what "realistically" means in this case. Are the simulated current strengths within a certain range of the observed ones?

*II. 235-237:* It is not clear to me how the different percentages relate. 15% on line 235, 10-20% on line 236 and 20% on line 237.

*I. 241:* As in line 219 (and following), I suggest to add a bit more information on what "reasonably similar" means. Some numbers would be beneficial to allow reproducibility and comparison with other studies.

*I. 265:* "south of 20°S" is rather unspecific. The EAC and EAUC sytems and the STCC are also south of 20°S. Additionally, there are also regions south of 20°S, where the intrinsic interannual variability is lower.

*II. 295-297:* There might be some answers (or hints) to these questions in Oliver and Holbrook (2014) and Bull et al. (2017). I agree though, that a thorough testing of this hypothesis should not be undertaken in this study.

*I. 319:* How is "veering eastward" defined?.

**6. Discussion and conclusion**

*II. 463-465:* You should note that the two studies (Travis and Qiu, 2017 and Rieck et al., 2018) investigated different regions. Travis and Qiu (2017) investigated a region from 165°E - 130°W which is much larger than the region used to investigate the STCC in this study. Averaging over such a large region should automatically lead to a smaller impact of intrinsic variability, as noted on lines 482-483. Rieck et al. (2018) investigated a region from 175°W - 153°W, which is almost entirely outside the region investigated here. You should better justify, why a comparison of this study with Travis

and Qiu (2017) and Rieck et al. (2018) is nonetheless valid. Penduff et al. (2011), Sérazin et al. (2015) and Rieck et al. (2018) all show that the ratio of intrinsic to total variability is not zonally uniform.

*ll. 472-473:* Given your filtering strategy to confine the analyzed variability to interannual time scales of 1 - 9 years (ll.142-143), it is surprising that you state to have found a link on decadal time scales.

**Technical corrections**

*1. Introduction*

*l. 25:* Instead of "[...], differently for different oceanic depths." I suggest to write something like "[...] with different impacts at different oceanic depths."
*ll. 29-30:* It should either be "Low-Latitude Western Boundary Currents" or "LLWBC".
*l. 36:* currents'
*l. 37:* masses'
*l. 54:* For better readability, I suggest to move "accordingly" to the end of the sentence.
*l. 74 "imprint":* Should be either "impact" or "imprint on".
*l. 111 "hampers":* Should be "hamper".

*2. Data, model description, methods*

*l. 162 "low-ass":* Should be "low-pass".
*ll. 186-197:* The NCJ, SCJ and Tasman Front are all three said to be labelled 3 on Figure 1a. The labels mentioned here do not agree with the labels in Table 1. The label 8 on Figure 1a is not described here. Given that the discussion quite prominently features the STCC, I suggest to add a section describing the STCC here, which should also be presented in Fig. 3.

**3. Deterministic versus chaotic oceanic transport variability**

*l. 232:* No comma after "(not shown)".
*l. 235:* I do not see dots in Fig. 1. Maybe there is a problem with the figure?.
*l. 238:* "EAC's and Tasman Front's".
*l. 251:* See comment to line 235.
*l. 256:* I suggest writing "of the ensemble mean 0-1000m zonal transport".
*l. 257:* no comma after "atmospherically-forced".
*l. 317:* I suggest using either "isoline" or "contour", not both.
*l. 321:* "eddies' ".

**4. Drivers of deterministic variability**

*l. 373:* Tchilibou et al. (2020) is not in the references.

**5. Spatio-temporal structure of the chaotic oceanic variability**

*ll. 400-401:* "imprint the transport" should be "impact the transports' ".
*l. 402:* I suggest deleting "hints of".
*l. 407:* I suggest writing "computed first" instead of "first computed".
*l. 407-410:* Aren't these two sentences describing the same thing?.
*l. 423:* "consists in" should be "consists of".
*l. 424:* I suggest writing "first two EOFs" instead of "two first EOFs".
*l. 432:* "behaves" should be "behave".
*l. 433:* "shows" should be "show".

**6. Discussion and conclusion**

[Figure]

*l. 446:* "than the deterministic atmospheric variability" should be at the end of the sentence.

*l. 457:* "varies" should be "vary".

*l. 467-468:* "density anomalies remotely forced" should be "remotely forced density anomalies".

*l. 468:* "EKE" should be "EKE's".

*l. 469:* "authors" should be "authors' ".

*Author contributions:* "run the experiments" should be "ran the experiments".

**Figures**

*Fig. 3:* The figure would benefit from a title (just for the whole figure, not for each panel), so the reader can see what this figure is about at the first glance. Additionally, at least the y-axes should get a unit.

*Fig. 4:* Panel d) lacks units for the colorbar.

*Fig. 5:* Panel a) lacks a colorbar.

*Fig. 11:* Units are missing.

**References**

**Bull**, C. Y. S., A. E. Kiss, N. C. Jourdain M. H. England and E. van Sebille: Wind forced variability in eddy formation, eddy shedding, and the separation of the East Australian Current. Journal of Geophysical Research: Oceans, 122, 9980–9998, doi:10.1002/2017JC013311, 2017.

**Oliver**, E. C. J., and N. J. Holbrook: Extending our understanding of South Pacific gyre "spin-up": Modeling the East Australian Current in a future climate, J. Geophys. Res.

Oceans, 119, 2788–2805, doi:10.1002/2013JC009591, 2014.

**Penduff**, T., M. Juza, B. Barnier, J. Zika, W. Dewar, A.-M. Tréguier, J.-M. Molines and N. Audiffren: Sea Level Expression of Intrinsic and Forced Ocean Variabilities at Interannual Time Scales. Journal of Climate, 24, 5652-5670, 2011.

**Sérazin**, G., T. Penduff, S. Grégorio, B. Barnier, J.-M. Molines and L. Terray: Intrinsic Variability of Sea Level from Global 1/12° Ocean Simulations: Spatiotemporal Scales. Journal of Climate, 28, 4279-4292, 2015.

---

## Referee Comment (RC2) · Anonymous Referee #2 · 30 Nov 2020

This paper described results from an ensemble of model simulations for the Southwest Pacific Ocean. The experiment design is very sensible, and clearly described. Readers might appreciate a few more details of how the ensemble was set up, but this is easily addressed. The authors analyse results from their ensemble to estimate how much interannual variability can be attributed to chaotic processes. They find that this can be 40-60% in some regions. This is higher than I expected. I wonder if there is a subtlety to their ensemble that needs to be considered. Specifically, I wonder whether there is a phase difference of interannual signals could be introduced between ensemble members – owing to the different initial conditions – that could explain some of the differences they attribute to chaotic processes. The authors go some way to look at

this with their analysis, but I think it would be worth looking at this before the paper is finalised. I expect that even if this is a factor, this study will be well worth publishing. It's very thought-provoking, and helps me think a bit differently about the circulation of this region. Some specific comments follow.

Re: ensemble perturbations

Perhaps the readers would be grateful for a bit more information on the perturbations to the initial ensemble.

Re: separation of interannual and chaotic variability

According to equations (1), all deviations from the time-varying ensemble mean are considered part of the chaotic ocean variability. But I wonder whether there could be some phase differences between members that are deterministic and unrelated to chaotic signals. Perhaps the different initial conditions could have some influence on the timing of interannual changes. Perhaps that interannual variability is equivalent, but just offset by some phase. Using the calculations outlined in section 2.2, I suspect these would be wrongly associated with chaotic variability.

I wonder if this could be checked by calculating the auto-correlation of transports, for example, at a few key locations to see if there is simply a phase-lag. Calculation of the coherence-squared and phase of the spectra may also help see whether this is a factor.

The EOF analysis (Figure 9 and 10) could perhaps be extended to look at this. Maybe you could look closely at the PCs of modes that are analogous between members. Does this show any offset in phase?

Maybe the authors would regard a shift in phase of an interannual signal as evidence of a chaotic process. If that's the case, I'm not sure I fully agree. Perhaps this could be more fully discussed in the paper.

Re: definition of transports

The term, "transport" is used to describe the "0-1000m integrated zonal and meridional transports ... computed from monthly mean velocities". I presume the velocities are integrated over depth, yielding units of mˆ2/s. This is consistent with the units in Figure 1 (mˆ2/s). I would be happy to see this stated explicitly.

This is a slightly unusual variable. It means that for the same "transport" value, points in coarser regions (eg at lower latitudes – at least for meridional transports) the volume transport is greater.

Is there a reason why the volume transports are not used? These would simply require the multiplication of the zonal or meridional grid spacing, yielding units of mˆ3/s.

---

## Author Comment (AC1) · 2 Jan 2021

General comments This study investigates the transport variability in the Southwest Pacific on interannual time scales with the help of the 1/4◦, 50-member ensemble simulation OCCIPUT. The focus is put on the influence of intrinsic (chaotic) oceanic variability on the observed transports in contrast to the deterministic variability forced by the atmosphere. The study also points out a general need for ensemble simulations to better quantify oceanic variability on interannual time scales, and includes a valuable discussion of the study's implications for the interpretation of e.g. observations. One important scientific value of this study is the validation or falsification of some results

that have already been published previously, but were based on single-member simulations or point-observations. If these results can (cannot) be reproduced with the OCCIPUT ensemble, they are much more (less) significant, due to the smaller errors using an ensemble and the advantage of estimating the chaotic variability that emerges internally in the ocean. The study is well-structured and comprehensive, and mostly needs some clarifications and minor technical revisions as listed below in the "Specific comments" and "Technical Comments".

We thank Jan Klaus Rieck for his encouragements, his very careful reading and his useful suggestions that helped to improve the manuscript. We agree with almost all his suggestions, and took into account all his comments.

However, there are three issues, that require some attention: 1. In the introduction, the latitude band 15-30◦S is identified as a region where many processes are not well understood and the OCCIPUT ensemble can help clarify. However, the current systems outside this latitude band (where previous studies reached more consistent conclusions) also recieve substantial attention in this mansucript. I therefore recommend, to more clearly point out that the validation of previous results with this novel modeling approach is an important aspect of this study, and not only the investigation of processes that could not be explained satisfacatorily so far.

We agree with this point. Indeed, we also investigate the deterministic versus chaotic variability equatorward of 15°S, and poleward of 30°S. We added this point in the introduction: "The results obtained with this novel modeling approach will help to understand the processes governing the interannual variability in the not well documented 15°S-30°S area, and will allow revisiting previous studies conclusions on transport and EKE deterministic forcing."

2. Another issue that needs some more work is the discussion about the results of this study in contrast to Travis and Qiu (2017) and Rieck et al. (2018). While I have no doubt that the general conclusion you draw is valid and the interannual variability in

the STCC region is mostly intrinsic, there are two points that should be clarified: - The different regions used in this study compared to Travis and Qiu (2017) and Rieck et al. (2018). Given that the simulations are global, you could easily extend the region you use to define the STCC and thus make it better comparable to the regions used in the other two studies. (Also see specific comment ll. 463-465 below) - While Travis and Qiu (2017) and Rieck et al. (2018) investigate decadal variability, this study focusses on interannual variability. This, and the implications, should be made more clear. This is true, indeed. The simulation is global, but extracting all the fields for extending the analyses is far from straightforward. This, and performing again all the computations and analyses would take a significant amount of time. Instead, we acknowledge these differences in the revised manuscript, and clearly state that our results are not immediately comparable.

We reformulated the discussion as follows: "Our results on the origin of the EKE variability (dominantly intrinsic) are in contrast with the findings of Rieck et al. (2018), who concluded that the EKE decadal variability in the STCC region was, uniquely among the subtropical gyres, mostly deterministic, driven by wind-forced decadal changes. In agreement with Travis and Qiu (2017) analyses of observations, they explained these decadal EKE modulation by decadal changes in vertical shear, driven by wind stress curl changes in the South Pacific, while still recognizing that other processes such as changes in stratification or remotely forced density anomalies also contribute to modulate the EKE's low-frequency changes. It is worth mentioning that our analyses are not directly comparable: firstly, the regions considered in these two studies differ from our region of focus here, and are much larger, extending eastward to the central part of the basin. Yet, Rieck et al. (2018) found a mostly deterministic EKE variance also in the 22°S-28°S to 160°E-180° region considered here, as can be seen in their Figure 2. Secondly, Rieck et al. (2018) studied the EKE decadal variability, whereas we focus on the interannual timescales. As we go toward larger spatial and temporal scales, it is probable that the intrinsic contribution to the total variance would diminish. It is also worth noting that we adopted here a different modeling strategy. Rieck et al. (2018)

used a long climatological run to study the sole imprint of intrinsic interannual variability. Here, we use 50 members with fully varying atmospheric forcing, which gives access to both intrinsic and atmospherically driven interannual oceanic fluctuations. Fully understanding the reasons behind our different conclusions on the EKE variability nature would require more analyses, which are beyond the scope of this paper." We hope this is now more rigorous and will satisfy the reviewer.

3. Additionally, the manuscript would benefit from an overview map. I suggest adding a new, large Fig. 1 to the manuscript, showing the bathymetry of the Southwest Pacific and the locations of the different seas (e.g. Coral Sea, Solomon Sea, etc ...), currents, and islands. We agree, that's a good suggestion that will help the readers not familiar with the region. We added this Figure in the paper, as Figure 1.

Specific comments 1. Introduction l. 30 "equatorial band": Do you mean the current bands? We meant the equatorial current system. This is now corrected.

l. 38 "These currents": Please specify which currents are referred to here; is the statement valid for all currents mentioned in the paragraph above? Done: it now reads "the currents in the whole Southwest Pacific"

l. 48 "wind anomalies in the southern hemisphere": Are these wind anomalies related to the SAM as well? Wind anomalies in the area (wind stress curl changes) have been suggested to be related to the Southern Annular Mode [Cai, 2006; Roemmich et al., 2007], or to decadal ENSO variations in the subtropics [Holbrook et al., 2005a, 2005b; Sasaki et al., 2008; Holbrook et al., 2011]. So it is not well known, and these different suggestions indicate that the changes in the region still need investigation. l. 57 "[...] south of 20âŮȩS [...]: In the paragraph above, ENSO is said to affect wind stress curl to 30âŮȩS. Here however, ENSO's influence is said to be restricted to North of 20âŮȩS. This is inconsistent. Thanks for pointing this apparent inconsistency. To be clearer, we modified the latitude in both sentences as "25°S", limit for the ENSO wind stress curl anomalies and ENSO's influence. In fact, there is no clear latitudinal limit,

as the wind curl anomalies decrease regularly poleward, and the Rossby waves phase propagation increase poleward, regularly decreasing the impact of ENSO basin-scale anomalies on the western transports. We hope this is now clearer. l. 64 "Between these two latitudes [...]: Does this refer to 15-30◦S? If yes, there is no need for this first part of the sentence. Agree; this is removed.

2. Data, model description, methods l. 130: I suggest to include the discussion about the impact of using a coupled system here (ll. 521-527). It is an important discussion but it does not fit at the end of the manuscript in my opinion.

Thanks for this suggestion. However, we consider that the important perspective of intrinsic variability in a coupled system fits well at the end of the manuscript. This is a suggestion for further studies, and we preferred to keep it where it was.

l. 131: Is there any specific reason to restrict the analyses to the period 1980-2015? Does the ensemble need the 20 years for the solutions to sufficiently diverge?

Thanks for this question. We now give explanation for this choice as follows: " We focus our analyses on the 1980-2015 period; before 1980 indeed, the buoyancy fluxes derived from the DFS5.2 forcing are devoid of interannual variability. Starting our analyses in 1980 thus yields an effective spinup time of 41 years within each member."

l. 131: How is the PDO index defined? (in fact line 215). We added the description of the PDO index: the leading principal component of North Pacific monthly sea surface temperature variability (poleward of 20°N for the 1900-93 period)

l. 144-145: Do you use the 180-day low-pass filtered velocities as the mean in the EKE calculations? Or as the deviations from that mean? This is not clear.

We use the 5-days velocities, filtered at high frequency by removing signals at frequencies lower than 180 days. Thus, we use the 180-day low-pass filtered velocities as the mean in the EKE calculations. We tried to make that clearer in the text.

3. Deterministic versus chaotic oceanic transport variability l. 219: I suggest to better

specifiy what "realistically" means in this case. Are the simulated current strengths within a certain range of the observed ones?

We added that in terms of latitudinal and longitudinal extensions, and in terms of mean transports, most currents are realistically simulated. We also added more details in the text referring to the description of the various currents, and compared the simulated mean transports to those observed from gliders, sections, or cruises.

ll. 235-237: It is not clear to me how the different percentages relate. 15% on line 235, 10-20% on line 236 and 20% on line 237.

Only areas where the spread is greater than 15% of the mean are shaded with dots (line 235). The other values are given to give more details, but cannot be inferred from the Figure. This is now stated by adding "not shown".

l. 241: As in line 219 (and following), I suggest to add a bit more information on what "reasonably similar" means. Some numbers would be beneficial to allow reproducibility and comparison with other studies.

Absolutely. We agree and added some EKE values, and provided more details about the model deficiencies.

l. 265: "south of 20◦S" is rather unspecific. The EAC and EAUC sytems and the STCC are also south of 20◦S. Additionally, there are also regions south of 20◦S, where the intrinsic interannual variability is lower.

Yes. "South of 20°S" is now removed.

ll. 295-297: There might be some answers (or hints) to these questions in Oliver and Holbrook (2014) and Bull et al. (2017). I agree though, that a thorough testing of this hypothesis should not be undertaken in this study.

We thank the reviewer for these suggestions. Although we knew about the Oliver and Holbrook paper, the interesting Bull et al. study did escape our research. We added

these papers in the references, and added sentences reflecting their conclusions.

l. 319: How is "veering eastward" defined?. We added "(ie, when the SSH isoline longitude starts to increase)", hoping this is now clearer.

6. Discussion and conclusion ll. 463-465: You should note that the two studies (Travis and Qiu, 2017 and Rieck et al., 2018) investigated different regions. Travis and Qiu (2017) investigated a re- gion from 165◦E - 130◦W which is much larger than the region used to investigate the STCC in this study. Averaging over such a large region should automatically lead to a smaller impact of intrinsic variability, as noted on lines 482-483. Rieck et al. (2018) investigated a region from 175◦W - 153◦W, which is almost entirely outside the region investigated here. You should better justify, why a comparison of this study with Travis and Qiu (2017) and Rieck et al. (2018) is nonetheless valid. Penduff et al. (2011), SeÌĄrazin et al. (2015) and Rieck et al. (2018) all show that the ratio of intrinsic to total variability is not zonally uniform.

Yes, this is true. Travis and Qiu (2017) and Rieck et al. (2018) focused on different regions, larger and further east. Yet, a careful look at their Figures (Figure 2b of Rieck et al., 2018) reveal that the ratio of EKE intrinsic variance on total EKE variance they found significantly differ from ours (see our Figure 4c). See also above how we refor- mulated the discussion.

ll. 472-473: Given your filtering strategy to confine the analyzed variability to interan- nual time scales of 1 - 9 years (ll.142-143), it is surprising that you state to have found a link on decadal time scales.

Thanks. We changed to "interannual".

Technical corrections 1. Introduction l. 25: Instead of "[...], differently for different oceanic depths." I suggest to write some- thing like "[...] with different impacts at different oceanic depths."

The term Âń impact" did not seem appropriate here; we changed to "with different

connectivity for different oceanic depths"

ll. 29-30: It should either be "Low-Latitude Western Boundary Currents" or "LLWBC". l. 36: currents' Thanks. Corrected.

l. 37: masses' Thanks. Corrected.

l. 54: For better readability, I suggest to move "accordingly" to the end of the sentence. Done

l. 74 "imprint": Should be either "impact" or "imprint on". Corrected, thanks. l. 111 "hampers": Should be "hamper". Corrected, thanks. 2. Data, model description, methods l. 162 "low-ass": Should be "low-pass". Thanks!

ll. 186-197: The NCJ, SCJ and Tasman Front are all three said to be labelled 3 on Figure 1a. The labels mentioned here do not agree with the labels in Table 1. The label 8 on Figure 1a is not described here. Given that the discussion quite prominently features the STCC, I suggest to add a section describing the STCC here, which should also be presented in Fig. 3.

Many thanks for pointing these errors. These are corrected. The STCC is a broad flow, composed of various branches. Defining its transport is thus complex, and we preferred not to isolate it as a specific key transport.

3. Deterministic versus chaotic oceanic transport variability l. 232: No comma after "(not shown)". Corrected l. 235: I do not see dots in Fig. 1. Maybe there is a problem with the figure?. We corrected the mistake: it should refer to Figure 1a and not 1b. We changed also "small black circles", hoping it is clearer. On our version at least these circles appear very clearly.

l. 238: "EAC's and Tasman Front's". Corrected l. 251: See comment to line 235. Quite surprising! They are even more clear in this Figure. . .

l. 256: I suggest writing "of the ensemble mean 0-1000m zonal transport". l. 257:

no comma after "atmospherically-forced". l. 317: I suggest using either "isoline" or "contour", not both. l. 321: "eddies' ". All done

4. Drivers of deterministic variability l. 373: Tchilibou et al. (2020) is not in the references. Thanks! Added.

5. Spatio-temporal structure of the chaotic oceanic variability ll. 400-401: "imprint the transport" should be "impact the transports' ". l. 402: I suggest deleting "hints of". Done, thank you. l. 407: I suggest writing "computed first" instead of "first computed". We removed "first". l. 407-410: Aren't these two sentences describing the same thing?. No. This is now better explained: first, EOF are applied on individual members. Then, on a combined 50 members together. l. 423: "consists in" should be "consists of". l. 424: I suggest writing "first two EOFs" instead of "two first EOFs". l. 432: "behaves" should be "behave". l. 433: "shows" should be "show".

All done, thank you.

6. Discussion and conclusion l. 446: "than the deterministic atmospheric variability" should be at the end of the sentence. l. 457: "varies" should be "vary". l. 467-468: "density anomalies remotely forced" should be "remotely forced density anomalies". l. 468: "EKE" should be "EKE's". l. 469: "authors" should be "authors' ". All corrected, thank you.

Author contributions: "run the experiments" should be "ran the experiments". Figures Fig. 3: The figure would benefit from a title (just for the whole figure, not for each panel), so the reader can see what this figure is about at the first glance. Additionally, at least the y-axes should get a unit. Fig. 4: Panel d) lacks units for the colorbar. Thanks for pointing this. Units were initially cm2/s2 per the total period (36 years). Ti ease comparisons with other studies, with changed to cm2/s2 per year. Fig. 5: Panel a) lacks a colorbar. Fig. 11: Units are missing.

All the figures have been corrected accordingly.

---

## Author Comment (AC2) · 2 Jan 2021

This paper described results from an ensemble of model simulations for the Southwest Pacific Ocean. The experiment design is very sensible, and clearly described. Readers might appreciate a few more details of how the ensemble was set up, but this is easily addressed. The authors analyse results from their ensemble to estimate how much interannual variability can be attributed to chaotic processes. They find that this can be 40-60% in some regions. This is higher than I expected. I wonder if there is a subtlety to their ensemble that needs to be considered. Specifically, I wonder whether there is a phase difference of interannual signals could be introduced between ensemble

members – owing to the different initial conditions – that could explain some of the differences they attribute to chaotic processes. The authors go some way to look at this with their analysis, but I think it would be worth looking at this before the paper is finalised. I expect that even if this is a factor, this study will be well worth publishing. It's very thought-provoking, and helps me think a bit differently about the circulation of this region. Some specific comments follow.

We thank the reviewer for his/her comments, that helped to improve the manuscript.

Re: ensemble perturbations Perhaps the readers would be grateful for a bit more information on the perturbations to the initial ensemble.

Yes, this is done. The draft mentioned: "the 50 members of the ensemble are generated in 1960 by activating a small stochastic perturbation in the equation of state within each member [Brankart et al. 2015; Bessières et al. 2017]. This perturbation is only applied for one year: it is switched off at the end of 1960, when the 50 members are restarted from slightly perturbed initial conditions and driven by the same atmospheric forcing." We changed to: "the 50 members of the ensemble are generated in 1960 by activating a small stochastic perturbation in the equation of state within each member (Brankart et al. 2015; Bessières et al. 2017). The small perturbations simulate the unresolved fluctuations of potential temperature and salinity. These fluctuations are generated using random walks [see Brankart et al., 2015 for details]. The initial perturbations are applied within each member for only one year in 1960: they are purely stochastic. The differences that grow between the members are therefore random by construction."

Re: separation of interannual and chaotic variability According to equations (1), all deviations from the time-varying ensemble mean are considered part of the chaotic ocean variability. But I wonder whether there could be some phase differences between members that are deterministic and unrelated to chaotic signals. Perhaps the different initial conditions could have some influence on the timing of interannual changes. Perhaps OSD
that interannual variability is equivalent, but just offset by some phase. Using the calculations outlined in section 2.2, I suspect these would be wrongly associated with chaotic variability. I wonder if this could be checked by calculating the auto-correlation of transports, for example, at a few key locations to see if there is simply a phaselag. Calculation of the coherence-squared and phase of the spectra may also help see whether this is a factor. The EOF analysis (Figure 9 and 10) could perhaps be extended to look at this. Maybe you could look closely at the PCs of modes that are analogous between members. Does this show any offset in phase? Maybe the authors would regard a shift in phase of an interannual signal as evidence of a chaotic process. If that's the case, I'm not sure I fully agree. Perhaps this could be more fully discussed in the paper.

We thank the reviewer for asking this important question. The small initial perturbations that are applied within each member during year 1960 are not just differences in phase: they are purely stochastic. These small perturbations are applied on the equation of state, and these random perturbations impact the geostrophic currents at the grid scale. Intrinsic variability is by construction seeded by small random fluctuations, which remain out-of-phase among the members and out-of-phase with the prescribed atmospheric variability throughout the run.

These small random velocity perturbations then progressively grow in amplitude due to oceanic non-linearities, generating the emergence of out-of-phase mesoscale perturbations among the members. As the ensemble spread tends to saturate in amplitude, the spatial and temporal scales of the intrinsic variability continue to grow (not shown) through non-linear inverse cascades of kinetic energy (Arbic et al. 2014; Sérazin et al (2018)). These are presumably the main processes that feed the interannual (small and large scale) intrinsic anomalies that we investigate.

To show that the transports in the various members are not just "offset by some phase", we computed the lagged correlations between low-frequency intrinsic variabilities of specific transports (time series in any member of this transport minus its ensemble
mean) within members i and j, with i=1-50 and j>i. For each couple (i,j) of members, we picked the maximum correlation C(i,j) with the associated lag lag(i,j). We thus obtained 49\*50/2=1225 pairs of lag and correlations for these transports; as an example, we show the results for the SCJ 0-1000m transport as a scatterplot in the Figure below.

If intrinsic signals were just offset by some constant phase, one would find a lag for each (i,j) couple that would correspond to large (and significant) values of correlations C. If intrinsic signals actually have time-varying random phase differences, then the C values would be randomly distributed, presumably around zero.

Figures R1: scatterplot between the lag of the maximum correlation, and the maximum correlation between the interannual intrinsic variability of the SCJ transport in all ensemble members. Red points are significant at the 90% level. The red dashed line shows the average of the maximum correlation for all lags.

The results show that the intrinsic variabilities in certain couples of members are correlated, but that most members do not show a significant correlation, at any lag. The correlations seem randomly distributed, with widely and randomly distributed lags. It should be noted that we correlate two low-frequency timeseries with a limited number of degrees of freedom. We purposely pick the lags (from -24 months to 24 months) at which the correlations are different from zero, and that is why there are few occurrences of maximum correlation lower than 0.1. Also, some members do exhibit a significant correlation at 90% confidence, but this is far from systematic. This result shows that we can rule out a systematic shift in phase of the interannual signals. We added sentences on this in the text, in section 2.1, 5 and 6.

Re: definition of transports The term, "transport" is used to describe the "0-1000m integrated zonal and meridional transports . . . computed from monthly mean velocities". I presume the velocities are integrated over depth, yielding units of mËĘ2/s. This is consistent with the units in Figure 1 (mËĘ2/s). I would be happy to see this stated explicitly. Yes, Indeed. These are in fact vertically integrated currents. This is

OSD
now explicitly stated. "0-1000m integrated zonal and meridional currents are computed from monthly velocities for each member. We call these quantities "vertically integrated transport", and their unit is m2 s-1. Âż This is a slightly unusual variable. It means that for the same "transport" value, points in coarser regions (eg at lower latitudes – at least for meridional transports) the volume transport is greater. Is there a reason why the volume transports are not used? These would simply require the multiplication of the zonal or meridional grid spacing, yielding units of mEE3/s.

This choice is made because it gives transports that are not dependent on the model grid size, and can thus be easily compared to observations (see Kessler and Cravatte, 2013) and other transports in models with  $\frac{1}{2}^{\circ}$  or  $1/12^{\circ}$  resolution, for example. Otherwise, if multiplied by the meridional or zonal grid spacing, it would be less easy to interpret. Transports in Sv (m3/s) are more meaningful for zonal and meridional sections, as shown in Figure 4 (previously Figure 3).
Fig. 1.

---

## Author Comment (AC3) · 2 Jan 2021

Following a suggestion from colleagues, we added in section and as a panel c in Figure 6 the timeseries of the bifurcation at the coast of Australia. We think indeed this is a key quantity, climatically relevant for the distribution of water masses equatorward and poleward. We found it to be dominantly deterministic, and correlated with ENSO. We hope it will be accepted as a valuable addition.

———————————————

---

## Author Comment (AC4) · 2 Jan 2021

Attached is the new Figure 1
* * *
[Figure]

**Fig. 1.**

---

## Author Response (AR2)

***Topic Editor Decision: Publish subject to minor revisions (review by editor)*** *(13 Jan 2021)*
*by Katsuro Katsumata*
*Comments to the Author:*
*The authors addressed appropriately all comments from both Reviewers 1 and 2.*

*I do not object the authors' decision to add Fig.6c and encourage authors to add the motivation ("... this is a key quantity, climatically relevant for the distribution of water masses equatorward and poleward." as stated in the authors' response) in the main text.*

*I would also ask authors to double-check the explanation of the term "bifurcation" at L.343 saying "...meridional transport along the coast of Australia changes from equtorward to poleward". The explanation sounds like the one for "retroflexion" (such as in Agulhas Current). Did you mean "zonal to meridional"?*

We thank Dr Katsumata for his comments and careful reading.
We followed his advice. Indeed, the definition of the bifurcation may have been misleading.
We are not studying the bifurcation of the western boundary current, but indeed the bifurcation of the incoming zonal SEC arriving at the coast of Australia, and partitioning into an equatorward and a poleward flow.
We changed the title of subsection 3.5. It now reads **"Variability of the EAC separation latitude and of the SEC bifurcation"**. We also modified our explanations of the bifurcation, and we hope it is now clearer:
"The latitude of the westward SEC bifurcation at the coast of Australia (ie the latitude separating waters that further flow into the equatorial system from those that feed the EAC) is an important quantity, climatically relevant for the distribution of water masses equatorward and poleward. This latitude, defined as the latitude along the coast of Australia at which the 0-1000m meridional transport changes from equatorward to poleward, is also computed…"
(lines 346-350)

*L.17: Expand EAC in Abstract.*
Done

*L.94: Please also expand "OCCIPUT" if this is an acronym in the 1st appearance.*
Done (lines 95-96)

*L.130: Brankart et al. (2015) is missing in the reference list.*
Added, thank you.

*L.483: "punctually" here is difficult to understand.*
We changed "punctually" to "locally".